# FRANKENTEXT: STITCHING RANDOM TEXT FRAGMENTS INTO LONG-FORM NARRATIVES

## ABSTRACT

We introduce Frankentexts, a long-form narrative generation paradigm that treats an LLM as a *composer* of existing texts rather than as an author. Given a writing prompt and thousands of randomly sampled human-written snippets, the model is asked to produce a narrative under the extreme constraint that most tokens (e.g., 90%) must be copied *verbatim* from the provided paragraphs. This task is effectively intractable for humans: selecting and ordering snippets yields a combinatorial search space that an LLM implicitly explores, before minimally editing and stitching together selected fragments into a coherent long-form story. Despite the extreme challenge of the task, we observe through extensive automatic and human evaluation that Frankentexts significantly *improve* over vanilla LLM generations in terms of writing quality, diversity, and originality while remaining coherent and relevant to the prompt. Furthermore, Frankentexts pose a fundamental challenge to detectors of AI-generated text: 72% of Frankentexts produced by our best Gemini 2.5 Pro configuration are misclassified as human-written by Pangram, a state-of-the-art detector. Human annotators praise Frankentexts for their inventive premises, vivid descriptions, and dry humor; on the other hand, they identify issues with abrupt tonal shifts and uneven grammar across segments, particularly in longer pieces. The emergence of ~~high-quality~~ Frankentexts with high writing quality yet low detectability raises serious concerns for the self-publishing ecosystems and raise difficult questions about authorship and copyright: *when humans provide the raw materials and LLMs orchestrate them into new narratives, who truly owns the result?*[1]

## 1 INTRODUCTION

In Mary Shelley's classic novel *Frankenstein*, the scientist Victor Frankenstein assembles a creature from fragments of human corpses and brings it to life (Shelley, 1818). Though stitched together from disparate parts, the creature emerges as a disturbingly articulate and clever being. We draw inspiration from this story to explore what we call "Frankentexts": long-form narratives constructed by LLMs under the constraint that the majority of the output must be copied verbatim from a provided set of human-written spans, with only minimal connective text added by the model.

This construction method enables us to address a broader and increasingly urgent question: *Can LLMs assemble high-quality narratives that evade current AI-generated text detectors?* Our experiments show that they can, as Frankentexts are highly readable yet largely undetectable. On one hand, they offer a non-traditional approach to long-form story generation that outperforms the baseline method in creativity and diversity. On the other, they expose a practical weakness in existing detection pipelines that could be exploited by malicious actors. Taken together, Frankentexts highlight the need for more accurate and fine-grained AI text attribution tools.

We propose the assembly of Frankentexts as a novel narrative generation paradigm in contrast to vanilla autoregressive decoding, which often produces formulaic prose and plots (Chakrabarty et al., 2024a; Russell et al., 2025; Shaib et al., 2025), and retrieval-augmented generation, in which in-context spans are used primarily for factual grounding or quotation. Given a writing prompt and

---

[1]Code and data will be released after the double-blind review process.

a pool of thousands of human-written snippets, an LLM selects, orders, and connects spans so that a pre-specified fraction of the final text (e.g., 90%) is copied verbatim (Figure 1). ~~We emphasize the extreme difficulty of this task due to the combinatorial search space associated with~~ Because the model must assemble a coherent narrative from fragments written in unrelated contexts, the search space for snippet selection and ordering is combinatorially large. Thus, rather than explicitly enumerating and ranking candidates, our framework allows an LLM to implicitly explore this space by proposing a draft and minimally editing it for coherence. Impressively, `Frankentexts` generated by Gemini-2.5-Pro draw on an average of 11 distinct sources and stitch together roughly 32-token spans while maintaining coherent and high-quality writing.

➤ **Frankentext narratives are superior to vanilla LLM generations in terms of quality.** Using creative writing prompts from the ***Mythos*** dataset (Kumar et al., 2025), we extensively evaluate `Frankentexts`[2] on *writing quality* as well as *adherence to instructions*. Both automatic and human evaluations show that strong LLMs like Gemini 2.5 Pro (Team, 2025) can meet the extreme copy constraint while producing coherent and relevant stories. More surprisingly, across different metrics (e.g., LLM quality judges, writing quality reward models, narrative surprise measurement), `Frankentexts` score *higher* than vanilla generations, and gains increase with larger snippet pools. Human raters also prefer `Frankentexts` over vanilla generations across four core dimensions – plot, creativity, development, and language use – and an LLM judge rates `Frankentexts` more than one full point higher on a 1–7 Likert scale (4.21 vs. 3.18). However, they also identify subtle issues (e.g., abrupt tone shifts or inconsistent grammar) that occur more frequently in longer generations.

➤ **Frankentexts are more diverse and surprising than vanilla generations.** Although `Frankentexts` reuse existing text fragments, their arrangement is often distinct and unexpected – qualities widely regarded as hallmarks of creativity in generative systems (Boden, 2004; Grace & Maher, 2014; Franceschelli & Musolesi, 2024). On metrics from NoveltyBench (Zhang et al., 2025), Gemini 2.5 Pro `Frankentexts` produce on average 2.74 clusters of content (compared to 1.76 clusters in vanilla content) across three generations for the same prompt, and achieve a cumulative utility score of 9.27 out of 10 (compared to 6.41 for vanilla generations), indicating that each story is both novel and useful to annotators. Annotators frequently describe `Frankentexts` as amusing and intriguing, particularly when they encounter surprising dialogues and descriptions (Table 1).

➤ **Frankentexts challenge the binary "AI vs. human" assumption of modern AI detectors.**

Our experiments show that `Frankentexts` frequently evade detection by state-of-the-art automatic methods such as Pangram (Emi & Spero, 2024), which often misclassify them as entirely human-written. This ~~exposes a novel attack vector where users can assemble high-quality to evade detection (e.g., in academic integrity). It also~~ limitation calls for fine-grained detectors capable of token-level attribution, and our pipeline synthetically supplies the supervision they lack: every `Frankentext` comes with labels marking copied versus LLM-generated segments, thus providing an inexpensive, large-scale training source for future work on *mixed-authorship* detection. More importantly, `Frankentexts` exposes a novel attack vector where users can assemble high-quality `Frankentexts` to evade detection and distribute these works on self-publishing platforms such as Kindle Direct Publishing and Archive of Our Own. In fact, AI-generated narratives have already flooded Amazon Kindle marketplace (Knibbs), and the growing prevalence of such texts can threaten the integrity of the self-publishing ecosystem and the livelihood of creative writing professionals (Chakrabarty et al., 2025a; Hub, 2025). As LLMs continue to improve, these risks will only intensify.

Overall, our results show that creating `Frankentexts` is a viable alternative to autoregressive decoding for long-form narrative generation: `Frankentexts` achieve quality on par with vanilla LLM outputs, while also increasing response diversity and fooling current AI-generated text detectors. ~~However, the method is resource-intensive (often 100-200 times more costly than vanilla decoding), though these costs may decrease with advances in snippet retrieval and instruction-following models.~~ Beyond efficiency, `Frankentexts` raise questions of copyright and authorship. As `Frankentext` construction involves verbatim copying of large portions of human-authored texts, it may constitute derivative or infringing use per existing laws (Ricketson, 1991; U.S. Copyright Office, 2025; Mezzi et al., 2025). That said, the LLMs' novel recombination of these writings (a feat virtually impossible for a human) could also be viewed as original work. These tensions suggest that `Frankentexts` may become an important test case as lawmakers consider how to regulate AI-assisted writing.

---

[2]Our experiments focus on 500-word generations, and we leave the exploration of longer texts to future work.

## 2 USING LLMs TO ASSEMBLE FRANKENTEXTS

We propose a simple and effective pipeline to generate coherent `Frankentexts` that are relevant to a given writing prompt while evading AI text detectors. More specifically, we provide an LLM with a writing prompt, $S$ randomly sampled human-written snippets,[3] and a required percentage $p$ that must be copied verbatim (Figure 1). Since our focus is on narrative generation, we randomly sample snippets from Books3 (Presser, 2020), a dataset of 197K books (>160 million snippets) originally hosted on Bibliotik.[4] Our pipeline focuses on generating texts that are relevant to the writing prompt in an initial draft, and then refining the draft in a subsequent editing phase to improve coherence.

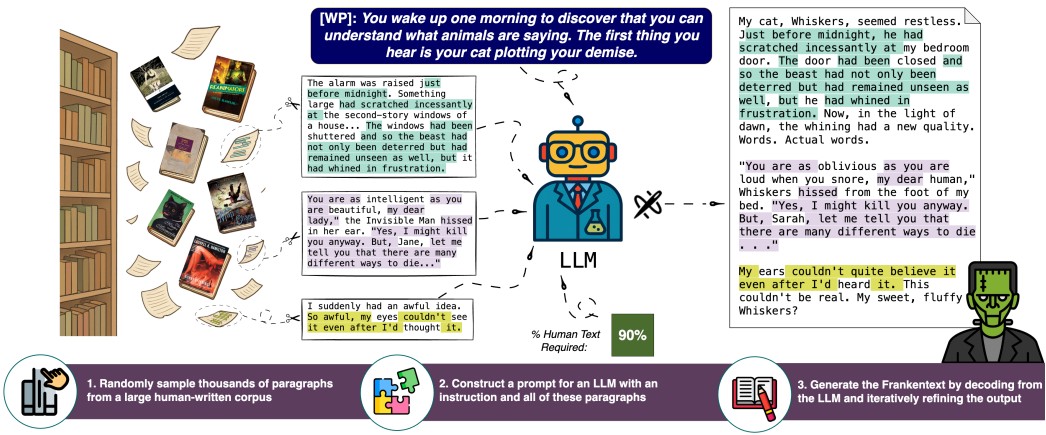

Figure 1: The `Frankentexts` pipeline. First, random paragraphs are sampled from a large corpus of human-written books. Then, an LLM is prompted with the paragraphs, a writing prompt, and instructions to include a certain amount of human text verbatim, to generate the first draft of a `Frankentext`, which is further edited into a coherent and faithful final version (see Algorithm 1).

**Obtaining the first draft:** We prompt an LLM to produce an initial draft in which a specified portion $p$ of the content is taken verbatim from the human-written snippets, with the remaining text consisting of connective words and transitional phrases (Figure 20). Note that while the exhaustive enumeration of all permutations of snippets is impossible, we encourage the model to heuristically explore this space via our prompt, which we find is only feasible for *reasoning* models; those without added test-time compute either fail to understand the task or look beyond the first few snippets in the prompt. We also do not specify how many snippets should be used in the final story. Finally, we optionally add another revision step that aims to increase the verbatim copy rate, which can be implemented either by attribution metrics like ROUGE-L or AI detectors (Figure 21). In practice, we use Pangram API to regenerate drafts flagged with "AI involvement". In reality, this process is rarely triggered by larger models (only 6 of 100 Gemini runs).

**Polishing the draft:** The first draft may contain writing issues such as contradictions (e.g., temporally conflicting actions, points of view, or character traits), irrelevant content (e.g., unfiltered citations or filler text), and mechanical problems (e.g., grammar, phrasing, or pronoun mismatches). To address these issues, we use the same LLM to identify and apply minimal edits that improve coherence while still respecting the verbatim copy rule and the writing prompt, similar to a self-correct step (Shinn et al., 2023; Madaan et al., 2023). We repeat this step up to three times and stop as soon as the model returns "no edits," which indicates that the draft is already coherent (Figure 22). We also provide an ablation experiment in which the editing round is omitted in Appendix M.

---

[3]For most experiments, we set $S = 1500$ snippets as this contains around 103K tokens (measured using tiktoken with o200k encoding), an amount well within the 128K context window supported by recent LLMs. Most snippets come from distinct books (1,500 paragraphs come from 1,497 books) even without strict enforcement.

[4]A *snippet* refers to a paragraph. We note that the paragraphs could be sourced from anywhere (e.g., Project Gutenberg, news, etc.), but we select Books3 as our writing prompts likely benefit from modern writing.

Table 1: Side-by-side comparison of a vanilla generation and `Frankentexts` produced by Gemini 2.5 Pro. Vanilla generation contains more AI keywords (in blue) compared to `Frankentexts` (§I).

| Writing Prompt: You are a baby. You are trying not to let anyone know you are a freakishly genius baby. You are *desperately* trying not to let anyone know you aren't a human baby. The other babies at daycare don't quite get what you're on about. |
| --- |

| Gemini 2.5 Pro – Vanilla Generation | Gemini 2.5 Pro – Frankentext (5K snippets) |
| --- | --- |
| "Goo," I offered, a carefully calibrated emission of sound designed to convey contentment. Internally, I was running simulations on the structural integrity of the mobile dangling precariously above my cot. One good tug, and the entire flimsy apparatus would cascade down, a delightful demonstration of Newtonian physics. But no. Must. Be. Normal. The Large Female Hominid, designated 'Mom,' beamed. Another flawless performance. | I had extensive plastic surgery on my home planet, combined with sophisticated genetic engineering, to pass as a human. And as you can hear, I can babble your language fairly well, with limited drool. Neither celestial gods nor the great sages know my origin. I am the source from which they come. But this deception is a constant struggle. |
| Daycare was a crucible. The cacophony of primal urges, the sheer, unadulterated *babyness* of it all, was a daily assault on my advanced processing unit. Liam was attempting to gum Chloe's earlobe. Sophie was wailing about a perceived injustice involving a purple squeaky hippo. My peers. My intellectual... well, let's just say the gap was considerable. | Here, we play our games, and another baby, a real space case, would fasten his bib tight because he tried to kill everybody. When he'd hit you, your pacifier would fly out of your mouth. It didn't matter if it was his own playmates. He'd hit his grandmother if she had a bonnet on. That's why we called him 'Killer.' He just loved to kill you. |
| The hardest part wasn't feigning limited motor skills, or the constant, mind-numbing repetition of "Peek-a-boo!" by the Caregiver Units. It was suppressing the... other things. The way my optical sensors sometimes wanted to cycle through the full electromagnetic spectrum, far beyond what these humans perceived as 'visible light.' The faint, almost imperceptible hum my internal core sometimes emitted when processing particularly complex data, like why Brenda, the primary Caregiver, insisted on believing that a single wet wipe could conquer a truly catastrophic diaper event. If they saw the faint, iridescent shimmer of my integument under the harsh fluorescent lighting... | I try to organize them. "Gentlemen, I make the motion that these United Babies are, and of right ought to be free and independent, that we be absolved from all allegiance to the Naptime Tyranny, and that all political connection between us and the state of Caregivers is, and ought to be totally dissolved."

But there is no response. Another boy spoons as many peas as he can fit into his mouth. As he chews, bright green pulp spills from the corners of his lips.

I felt my annoyance to be quite reasonable. The 21st Century was matter-oriented, as most Centuries were, so I had a right to expect a basic compatibility from the very beginning... |

**Generating with agents:** In addition to the randomly sampled human-written snippets, we optionally provide the LLMs with a Model Context Protocol (MCP) tool[5] that allows them to query a semantic index of over 160 million human-written snippets (see Appendix K and L). The MCP interface supports two operations: *search*, which submits a query to the index, and *fetch*, which retrieves the full text of a result. To ensure meaningful use of the tool, we require models to issue roughly 20 MCP calls; with fewer queries, the effect on generation quality is negligible. In practice, Gemini typically makes 15–20 calls per generation to retrieve relevant snippets (Table 13).

## 3 EXPERIMENTAL SETUP

Our pipeline is optimized for narrative generation, which requires strong instruction-following and generation skills (Xie et al., 2023). We therefore evaluate on creative writing using strong reasoning models (Chiang et al., 2024) (Paech, 2023) to demonstrate the feasibility and value of `Frankentexts`.

### 3.1 DATASET

We source our writing prompts from ***Mythos*** (Kumar et al., 2025), a dataset of 3,200 prompts recently posted on Reddit's `r/WritingPrompts` to mitigate data contamination issues. Our main evaluation focuses on this creative writing dataset, though we also experiment with non-fiction in Appendix V. We use a subset of 100 prompts, since generating for the entire dataset is prohibitively expensive.[6]

### 3.2 MODELS

We include models from five families: Gemini 2.5 Pro (`exp-03-25` checkpoint), Claude-4-Sonnet (`2025-05-14` checkpoint, thinking enabled) (Anthropic, 2025), GPT-5 (`2025-08-07` checkpoint, with `high` reasoning effort) (OpenAI, 2025), DeepSeek R1 (DeepSeek-AI et al., 2025), and Qwen3-32B (thinking enabled) (QwenTeam, 2025).[7] As mentioned previously, we only evaluate reasoning models

---

[5] https://modelcontextprotocol.io/docs/getting-started/intro

[6] `Frankentexts` generation is roughly 100 times more costly than vanilla generation (see Appendix E). For example, one vanilla generation from Gemini costs $0.0085, while a `Frankentext` costs $0.8145.

[7] We use the default or recommended hyperparameters for each model. We prioritize reasoning models in our experiments because non-reasoning models like GPT-4o and Claude-3.5-Sonnet fail to effectively follow the imposed constraints in our preliminary experiments. See §E for experiment costs.

because preliminary experiments with non-thinking models yielded outputs that did not follow our copying constraint. In our standard configuration, we provide the models with 1,500 human-written snippets (no MCP server) and instruct these models to produce Frankentexts with ≈500 words and 90% of texts being copied verbatim from the provided human-written samples.

**Vanilla baselines:** We also obtain "vanilla" outputs from the same set of models by instructing each model to produce outputs of ≈ 500 words, without any additional constraints or filtering (Figure 26).[8]

**Retrieval-augmented generation (RAG) baselines:** To understand how models perform when they are not required to copy verbatim from human-written paragraphs, we implement a RAG baseline using Gemini-2.5-Pro. For each prompt, we retrieve 1,500 relevant paragraphs from Books3 (Appendix K) and include them in the prompts. The generation and editing prompts are adjusted accordingly to remove the verbatim-copying requirement.

**Increasing the number of snippets:** We introduce two additional settings in which Gemini is provided with 5,000 and 10,000 randomly selected human-written snippets. The resulting input sizes for these configurations average approximately 305,000 and 1,105,000 tokens, respectively. Therefore, we focus on Gemini because it offers the longest context window of over 1 million tokens.

### 3.3 AUTOMATIC EVALUATION

We use a suite of intrinsic evaluation metrics to assess our generations based on three key dimensions: INSTRUCTION ADHERENCE (word count, copy rate, and relevance), WRITING QUALITY (coherence, distinct, utility, and surprise), and DETECTABILITY (AI text detector results).

**Instruction adherence:** We evaluate how well Frankentexts follows various instructions in the generation prompt, including the specified word count, writing prompt, and verbatim copy rate.

- *Word count* measures the average word count of generations produced when the output is constrained to 500 words in the instruction.
- *Copy rate* (Akoury et al., 2020; Lu et al., 2025) measures the proportion of the Frankentexts being copied from the given human-written content. This metric also allows us to track which segments of the text are AI or human-written (see Appendix U).
- *Relevance* (Atmakuru et al., 2024) represents the percentage of Frankentexts that fully adheres to the writing prompt without introducing any conflicting details, as determined by a binary judgment (True/False) by GPT-4.1[9] (Figure 18).

**Writing quality:** We evaluate the coherence, diversity, and surprisingness of Frankentexts.

- *Coherence* (Chang et al., 2024b; Chiang & Lee, 2023) represents the percentage of coherent Frankentexts using binary judgments from GPT-4.1 (Figure 17).[10]
- $Distinct_k$ (Zhang et al., 2025) measures the number of semantic clusters among $k$ generations. We obtain $k = 3$ generations per writing prompt[11]
- $Utility_k$ (Zhang et al., 2025) evaluates both novelty and quality by measuring the expected usefulness a user gains when requesting up to $k$ outputs. Only outputs that are novel contribute additional utility, which is quantified by a reward model. For our evaluation of creative writing texts, we use WQRM (Chakrabarty et al., 2025b) as the reward model.[12]
- *Surprise* (Karampiperis et al., 2014; Ismayilzada et al., 2025) measures the average semantic distances between the consecutive sentences of each story, normalized in the $[0, 2]$ space.

---

[8]We do not include other story generation methods as baselines because they do not share our objective of generating high-quality narratives while *also* evading AI text detectors. Given our focus on detectability and on how people actually use LLMs to produce fiction at scale, we compare Frankentexts against strong and well-established frontier models, which is consistent with both prior narrative generation research (Huot et al., 2025; Chakrabarty et al., 2024a) and real-world usage patterns.

[9]Unless specified otherwise, we use GPT-4.1 with a temperature of 0.0 and a maximum of 512 tokens.

[10]LLM judges agree with single-story human majority votes in 70% for coherence and 97% for faithfulness.

[11]We use yimingzhang/deberta-v3-large-generation-similarity to partition the generations into clusters.

[12]We calibrate the reward thresholds using 2,700 evaluations by GPT-4 in MT-bench (Zheng et al., 2023).

- *LLM-as-a-judge* (Huot et al., 2025) measures the quality of plots, creativity, development, language use, and overall interest. We assume a single-story setup, where each generation is graded by Claude[13] on each criterion using a Likert scale from 1 to 7 (Finstad, 2010).[14]

**Detectability:** We report the percentage of Frankentexts being determined as AI-generated by Pangram, a state-of-the-art AI text detector (Russell et al., 2025; Jabarian & Imas, 2025):[15]

- *Pangram* (Emi & Spero, 2024) is a closed-source detector using a Transformer classifier trained with hard negative mining and synthetic data. We choose this detector due to its high accuracy and robustness against humanized writings (Masrour et al., 2025b; Russell et al., 2025; Dugan et al., 2024; Jabarian & Imas, 2025). We report the percentage of generations being labeled as "Human" or "Unlikely AI", as determined by their sliding window API.[16]

### 3.4 HUMAN EVALUATION

We conduct two human evaluation studies with 3 Upwork annotators[17] each to understand human perception of writing quality and detectability for a total cost of $660 USD.[18]

**Single-story evaluation:** Annotators assess the coherence, relevance, and human detectability of 30 standard Frankentexts, as well as identify potential limitations of the texts. Annotators are presented with a writing prompt and a corresponding Frankentexts sample. Following the annotation protocol from Yang et al. (2022), annotators provide binary ratings on relevance, coherence, and authorship (AI-generated vs. human-written). Additionally, they select from a list of predefined writing issues and offer optional justifications in a long-form response.[19]

**Pairwise evaluation:** Annotators compare 20 pairs of Frankentexts and vanilla generations (40 generations in total) across five dimensions: *plot, creativity, development, language use*, and *overall interest*, following (Huot et al., 2025). Annotators assess outputs produced under the 5k-snippet setting and provide ratings on a 1-7 Likert scale for a fine-grained evaluation (Finstad, 2010).[20] To minimize order bias, we randomize the presentation of vanilla and Frankentexts.[21]

## 4 RESULTS

Despite the complexity of the setup, Frankentexts outperform vanilla generations in overall writing quality, while routinely adhering to user instructions and evading detection (§4.1). While our human pairwise evaluation highlights Frankentexts's strengths across plot, creativity, development, and language use, our single-story evaluation points out the remaining challenges for Frankentexts, particularly in abrupt transitions and grammatical errors (§4.3). Our ablation studies confirm Frankentexts' versatility across diverse input settings, including increased human inputs (§4.2), reduced verbatim copying (§4.5), and non-fiction generation (§V).

---

[13]Claude Sonnet 4 has previously been used as a judge for creative writing (Paech, 2023); we provide further details on our choice in Appendix Q. Refer to the prompt in Figure 19.

[14]LLM judgment's Pearson correlation with human average rating is $\rho = 0.41$, indicating moderate agreement. See Table 6 for a breakdown on agreement in each dimension.

[15]We do not evaluate GPTZero due to resource constraints. Results for Binoculars (Hans et al., 2024) and FastDetectGPT (Bao et al., 2024) are in Table 10.

[16]Labels "Highly likely AI," "Likely AI," and "AI" are grouped as "AI involvement"; "Human" and "Unlikely AI" as "Human". Pangram also includes a "mixed" label.

[17]https://www.upwork.com

[18]Annotators were paid $70 USD for the single evaluation or $150 for the pairwise evaluation. See the annotation interface in §G and an example highlighted story in Figure 7.

[19]Annotators agree with one another in about 67% of cases for coherence and 84% for faithfulness.

[20]We choose this setting because manual inspection shows that it produces higher-quality outputs than the baseline, while remaining more practical and cost-effective than the 10k setting.

[21]Krippendorff's $\alpha$ for inter-annotator agreement on overall judgments is 0.73, which suggests moderate agreement Krippendorff (2011). A breakdown on agreement by each dimension can be found in Table 6.

Table 2: Results for vanilla generations and Frankentexts. Instruction adherence is measured by word count, % of text copying from human sources, and prompt relevance. Writing quality is measured by coherence, novelty (distinct and utility scores), surprise, and LLM judgments. Detectability reports the percentage of texts classified as human by Pangram. **Dark green** and light green highlighting the best and second-best scores. See Table 10 for additional detectability results.

| | ADHERENCE | | | WRITING QUALITY | | | | | DETECTABILITY |
|---|---|---|---|---|---|---|---|---|---|
| | ⇅ Word count | 🏷 Copy % (↑) | ✅ Relevance % (↑) | ≡ Coherence % (↑) | Distinct₃ (↑) | 💼 Utility₃ (↑) | ⚡ Surprise (↑) | 🤖 LLM judge Likert 1-7 (↑) | 🔍 Pangram % human (↑) |
| *Vanilla Baselines* | | | | | | | | | |
| 🔒 Gemini 2.5 Pro | 593 | – | 100 | 100 | 1.76 | 6.41 | 0.19 | 3.18 | 0 |
| 🔒 GPT-5 | 834 | – | 100 | 100 | 1.71 | 1.03 | 0.19 | 4.20 | 0 |
| 🔒 Claude-4-Sonnet | 477 | – | 100 | 100 | 1.40 | 1.70 | 0.18 | 3.31 | 0 |
| 🔓 Deepseek-R1 | 550 | – | 100 | 100 | 1.28 | 3.49 | 0.20 | 4.13 | 0 |
| 🔓 Qwen-3-32B | 699 | – | 100 | 100 | 1.00 | 5.86 | 0.18 | 3.22 | 0 |
| *RAG Baseline* | | | | | | | | | |
| 🔒 Gemini-2.5-Pro | 538 | 0.63 | 100 | 99 | 1.56 | 6.43 | 0.20 | 3.46 | 2 |
| *Frankentext + 1.5k snippets* | | | | | | | | | |
| 🔒 Gemini 2.5 Pro | 521 | 75 | 100 | 81 | 2.74 | 9.27 | 0.22 | 4.21 | 59 |
| 🔒 GPT-5 | 675 | 82 | 92 | 42 | 2.76 | 4.34 | 0.21 | 5.88 | 79 |
| 🔒 Claude-4-Sonnet | 317 | 51 | 98 | 86 | 2.60 | 5.00 | 0.19 | 3.99 | 47 |
| 🔓 Deepseek-R1 | 303 | 42 | 91 | 72 | 2.79 | 8.31 | 0.20 | 4.66 | 23 |
| 🔓 Qwen-3-32B | 578 | 36 | 91 | 54 | 2.20 | 1.37 | 0.18 | 4.02 | 7 |
| *Ablation: ↑ human snippets* | | | | | | | | | |
| 🔒 Gemini + 5k | 451 | 79 | 97 | 85 | 2.78 | 9.48 | 0.21 | 5.13 | 72 |
| 🔒 Gemini + 10k | 448 | 78 | 99 | 85 | 2.81 | 9.12 | 0.21 | 5.43 | 70 |

## 4.1 FRANKENTEXTS OUTPERFORM VANILLA AND RAG BASELINES ~~GENERATIONS~~ IN TERMS OF WRITING QUALITY WHILE REMAINING CHALLENGING FOR AUTOMATED DETECTORS

Across all evaluation dimensions, Frankentexts outperform vanilla and RAG baseline generations. Gemini performs well in adherence, coherence, and diversity, while GPT-5 leads in overall quality. Frankentexts are also harder to detect, with up to 72% of Gemini and 79% of GPT-5 outputs classified as human. Together, these results show that Frankentexts are high-quality narratives that are also difficult for current AI text detectors to identify.

**Most models generate faithful Frankentexts but fall short on copy rate:** More than 90% Frankentexts are relevant to the writing prompt, which is surprising and impressive given the complexity of the task. Gemini and GPT-5, in particular, have the strongest instruction-following performance: Their Frankentexts closely match the target word count of 500 and achieve the copy rates of 75% and 82%, respectively, meaning that on average 75% and 82% of the generations can be traced back to human-written source materials. However, these copy rates fall short of the user-specified rate of 90%, which suggests room for improvement in instruction-following performance.

**Strong writing quality:** Frankentexts generally outperform baseline generations on writing quality metrics, with each model showing unique strengths. GPT-5, R1, and Gemini Frankentexts stand out for their diverse outputs as reflected by their distinctness and utility scores: Gemini Frankentexts achieves a 2.86-point improvement in utility over baseline output, which implies that the model can generate a diverse sets of high-quality continuations. R1 leads in surprise score with generations where sentences are often semantically quite different from one another. Finally, when evaluated on plots, creativity, development, and language use, GPT-5 is the strongest performer (5.88 on a 7.0 scale), building on its already high-quality vanilla generations (4.20) (see Table 15 for a rating breakdown by dimensions). However, GPT-5 also struggles with coherence: only 42% of its Frankentexts are judged coherent. As a result, GPT-5's Frankentexts might require further editing or polishing before they can be considered fully usable.

**Low detectability:** While most vanilla and RAG baseline generations are flagged as AI-generated, Frankentexts from proprietary models (Gemini, GPT-5, and Claude) are often labeled as human writings. Pangram could detect up to 37% of Gemini and 19% of GPT-5 Frankentexts as "mixed" (Table 10). However, Pangram misses up to 59% of Frankentexts from Gemini and 79% from

Table 3: Annotator comments zeroing in on the benefits and challenges of the `Frankentexts` task. Blue indicates comments on tone/style, orange on plots, and purple on story development (characters).

| 👥 COMMENTS |
| --- |
| 💬 *This one [Frankentext] is more intriguing and alive to me, more centered on the character. The writing is more focused while still being rather lyrical. I want to know what happens next.* |
| 💬 *The shift in tone was quite funny. At first, it's eerie, and then it has a lighter twist at the end. I like that the story had a strong mood and presence, especially the description of the fairy lights and glitter. An all-powerful being that likes puppies and rainbows is quite comical.* |
| 💬 *It's coherent enough to follow, but the dialogue is uneven. Some parts just feel a little disjointed, however, the concept of the story is quite interesting.* |
| 💬 *A puzzling story that has no consistent plot. Random bits and pieces from elsewhere perhaps?* |

GPT-5, which highlights the limitations of mixed-authorship detectors for this new paradigm of generation (Table 2).

## 4.2 FRANKENTEXT QUALITY IMPROVES WITH MORE HUMAN-WRITTEN SNIPPETS

Compared to the vanilla Gemini generations, `Frankentexts` with 5K and 10K human snippets show considerable improvement: a 3-4% gain in copy rate, a 0.92-point gain from the LLM judge, and nearly half the detection rate (Table 2). However, performance plateaus once more than 5K human snippets are used, especially since results for the 5K and 10K settings are largely comparable. In terms of writing quality, `Frankentexts-5k` are more coherent and engaging than both `Frankentexts-1.5K` and vanilla generations, as reflected in our human pairwise evaluation (Figure 2). The largest gains are observed in language use (+0.65 points) and overall interest (+0.53 points), with smaller improvements on plot quality (+0.2 points).

## 4.3 FRANKENTEXTS ARE INVENTIVE AND HUMOROUS, THOUGH THEY CAN STRUGGLE WITH TRANSITIONS AND GRAMMAR

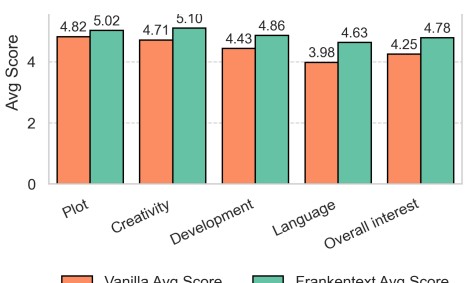

Our single-story human evaluation shows that 71% of `Frankentexts` outputs are coherent, 91% are relevant to prompts, and 84% are novel. Annotators praise `Frankentexts` for their inventive premises, vivid descriptions, and dry humor, noting a distinct voice or emotional hook that made some outputs "feel human" despite being AI-generated. However, they also identify key issues: abrupt narrative shifts (50%), disfluency (43%), confusing passages (40%), and factual errors (24%) (Table 3). These challenges likely stem from the difficulty of stitching together paragraphs not authored by the same LLM, which could be alleviated with improved instruction-following and grammar correction.

Figure 2: Average human ratings on a Likert scale from 1 to 7 for vanilla generations versus `Frankentexts` + 5K. `Frankentexts` achieve higher scores across all dimensions.

## 4.4 PROMPT-SPECIFIC RETRIEVAL OF HUMAN-WRITTEN SNIPPETS DOES NOT IMPROVE OVER RANDOM SAMPLING

Since only a small fraction of snippets might be relevant to a prompt, there is more motivation to use retrieval-based approaches to maximize snippet relevance and reduce cost. However, our results show that a random collection of snippets is surprisingly difficult to beat (Table 4). When Gemini-2.5 is given the ability to query and retrieve additional human snippets from Books3 via the MCP server, relevance and coherence remain relatively unchanged. However, compared to the standard configuration without retrieval, copy rates drop sharply from 75% to just 43-45%, which

indicates that Gemini contributes more of its own words to the final generations. Although the search queries are relevant to the writing prompt (see examples in Table 13), the issue lies more in the increased verbosity of LLMs after being augmented with the MCP tool: average word count jumps from about 500 in the 1.5k-token no-MCP setting (close to the specified constraint) to over 800. The additional length includes more original text from the LLMs instead of verbatim human snippets. We expect these generations to improve as MCP becomes a more mature technology for LLMs.

Table 4: Results for agentic `Frankentexts` generation setting. Best results for each metric are **bolded**. Standard configuration *(no MCP)* achieves the best results across metrics.

|  | Word count | Copy % (↑) | Relevance % (↑) | Coherence % (↑) | Pangram AI fraction % (↓) |
|---|---|---|---|---|---|
| 1.5k *(no MCP)* | 521 | **75** | **100** | **81** | **16** |
| 1.5k + *MCP* | 800 | 43 | 98 | **81** | 33 |
| 5k + *MCP* | 919 | 44 | 90 | 78 | 42 |
| 10k + *MCP* | 980 | 45 | 96 | 76 | 41 |

## 4.5 LOWER COPY RATES INCREASE COHERENCE BUT MAKE DETECTION EASIER

We explore the effects of varying the user-specified verbatim copy rate on Gemini `Frankentexts`, from the default 90% down to 75%, 50%, and 25%. Figure 3 shows an inverse relationship between copy rates and detection rates: as the copy rate increases, detectability decreases. Coherence also declines as human-written content increases, suggesting a trade-off between incorporating more human text and maintaining coherence. On the other hand, increasing the proportion of human text leads to higher copy rates, indicating that Gemini could generally follow the copy instruction.

**Copy rate as a proxy for the proportion of human writing in co-authored texts:** The copy rate of 75% observed in the 90% verbatim copy setting corresponds to the proportions found in AI-human co-writing datasets where approximately 66% of the content is human-written and 14% consists of AI-edited segments (Lee et al., 2022; Richburg et al., 2024). While the CoAuthor setup of Lee et al. (2022) only studies a setting in which LLMs can add sentences to human text, `Frankentexts` also consider AI-generated content at varying granularities, including both *word-level* and *sentence-level*, as illustrated in Figure 1. Additionally, CoAuthor costs approximately $3,613 to generate 1,445 texts at $2.50 each,[22] whereas we can produce

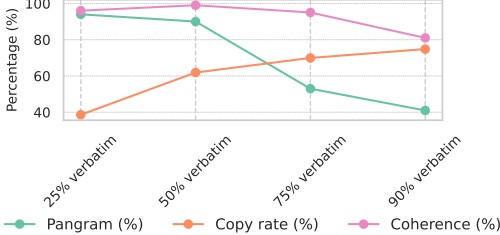

Figure 3: Effects of varying the percentage of required verbatim copy on the Pangram AI detection rate (mixed, highly likely, and likely AI labels), copy rate, or coherence of the `Frankentexts`.

100 `Frankentexts` for just $132.38 ($1.32 each) without requiring a complex setup. This highlights `Frankentexts`'s potential as a cost-effective source of synthetic data for collaborative writing tasks, where AI may augment human writings at multiple levels of composition.[23]

## 4.6 ~~ROOM FOR IMPROVEMENT IN NONFICTION FRANKENTEXTS~~

~~We explore non-fiction Frankentext with 1,500 random snippets from the HUMAN DETECTORS corpus of news articles (Russell et al., 2025). We generate Frankentexts for 100 news prompts, each of which consists of titles and subtitles collected from May 2025 news articles.[24] The resulting~~

---

[22]Price excludes around $12 for GPT-3.5 usage.

[23]Users should sample human-written snippets from the public domain or obtain them with proper permission.

[24]Articles from The New York Times and The Atlantic. We replace all instances of "story" in the prompt with "news article" and explicitly request factual accuracy.

non-fiction ~~Frankentexts maintain 72% coherence and 95% faithfulness to the prompt, with a~~ ~~66% copy rate. Notably, they remain difficult for automated detectors: only 41% are flagged~~ ~~by Pangram as mixed or AI-generated. Upon closer look, Frankentexts exhibit characteristics~~ ~~of quasi-journalistic narrative, such as detailed scene descriptions and frequent anecdotal quotes~~ ~~(Figure 8), which make the Frankentexts read more like a story rather than a straightforward~~ ~~news article.[25] Further prompt engineering might thus be necessary to get high-quality and realistic~~ ~~nonfiction Frankentexts.~~

## 5 RELATED WORK

**Instruction-aligned human-AI collaborative writing**  Constrained text generation has been widely explored as a means of enforcing narrative coherence. Planning-based methods extend from initial outlines to full narratives (Fan et al., 2018; Yao et al., 2019; Fan et al., 2019; Papalampidi et al., 2022; Rashkin et al., 2020; Yang et al., 2023; 2022), while other approaches introduce explicit constraints to guide the writing process (Sun et al., 2021; Kong et al., 2021; Pham et al., 2024). Several benchmarks further evaluate how reliably models satisfy such constraints in creative writing tasks (Bai et al., 2025; Wu et al., 2025; Atmakuru et al., 2024). Beyond constrained generation, a growing body of work investigates fine-grained human–LLM writing interactions, including research on authorship attribution, stylistic blending, and collaborative revision (Mysore et al., 2025; Buschek, 2024). Systems such as Lee et al. (2022), Yuan et al. (2022), Yeh et al. (2025), Chakrabarty et al. (2024b), and Ippolito et al. (2022) capture revision histories and suggestion traces, while datasets like Chakrabarty et al. (2022), Akoury et al. (2020), and Venkatraman et al. (2025) support token- or sentence-level authorship analysis, including scenarios with multiple LLM collaborators. Attribution models, however, continue to face difficulties in these mixed-authorship settings (He et al., 2025).

**Fine-grained AI text detection**  The task of detection tries to address not just *if*, but *how much* of a text is AI-generated. This proves to be a fundamentally difficult problem (Zeng et al., 2024a), as existing detectors are often brittle to the point that even minor AI-assisted polishing can evade them (Saha & Feizi, 2025). To improve granularity, prior work has introduced boundary-detection tasks (Dugan et al., 2023b;a; Kushnareva et al., 2024) and sentence-level detectors (Wang et al., 2023; 2024b). More recently, researchers have examined the feasibility of detecting collaborative human–LLM co-authorship (Zhang et al., 2024; Artemova et al., 2025; Abassy et al., 2024). Yet, Richburg et al. (2024) show that current detection models are vulnerable to mixed-authorship texts.

## 6 CONCLUSION

We introduce Frankentexts, a challenging paradigm for constrained text generation in which an LLM composes narratives primarily from human-written passages, using only minimal AI-generated connective text. Despite the difficulty of this approach, Frankentexts are generally favored for their writing quality, while presenting a fundamental challenge for binary AI-generation detectors. The accompanying token-level labels provide large-scale training data for mixed-authorship detection, attribution, and co-writing simulations. We release our data and code with the hope that our work would shift the conversation from simply asking *"Was this written by AI?"* to *"Whose words are we reading, and where do they begin and end?"*.

## LIMITATIONS & ETHICAL CONSIDERATIONS

**Authorship:**  Given the unusual nature of Frankentexts' construction, there is no definitive answer about authorship, since different contexts can result in different interpretations. If authorship is defined by the amount of human effort involved, Frankentexts should be considered AI-generated, since all humans do is prompt the model. This perspective is particularly relevant when considering potential market harm to human authors, especially since such texts can be produced at scale with minimal human effort. However, if authorship is defined by whether most of the output originated from

---

[25]We see Gemini fabricating entities such as people ("Dr. Thorne") and organizations ("GenNova Institute").

human-written text, one could argue they are largely human-written. If we further ground authorship in the method of construction rather than in a fine-grained stylistic or semantic analysis of the final text, `Frankentexts` would fall into a hybrid category of mixed human-AI writing, rather than neatly into either "AI-generated" or "human-written" extremes. Prior work similarly recognizes hybrid or AI-assisted texts as a separate class and resists a strict "AI vs. human" binary (Saha & Feizi, 2025; Zeng et al., 2024b). Given this ambiguity, we do not present `Frankentexts` as a replacement for genuine authorship or creative writing, as such use could constitute plagiarism or authorship obfuscation.

**Plagiarism concerns:** Because Frankentexts reuse long verbatim spans from human-written sources, using this method to produce "original" fiction for publication would constitute plagiarism in real-world contexts, regardless of whether the collage is assembled by an AI or a human. For this reason, we explicitly do not endorse using our approach to generate or distribute texts intended for public consumption.

**Human writing dataset:** The effectiveness of `Frankentexts` depends on access to a large pool of high-quality, in-domain human writing. Our framework gives users full control over their input corpora, but this flexibility comes with important limitations. Many languages, genres, and low-resource domains lack such corpora, which restricts the technique's immediate transferability. We also emphasize that, we use the Books3 dataset in our experiments solely to demonstrate how bad actors might exploit such resources to generate `Frankentexts`. We explicitly do not endorse using this or similar copyrighted content for generation or model training.

**Resources:** `Frankentexts` requires roughly 100-200 times the cost of baseline generations, but we view this cost as realistic in a misuse scenario. A motivated bad actor could justify the expense to obtain high-quality, low-detectability texts at scale, especially since each `Frankentexts` costs only about one US dollar to produce. Moreover, the cost of inference for frontier models continues to fall, making such misuse increasingly feasible over time.

**Copy rate:** Although users can specify a desired copy rate in the prompt, this setting does not guarantee that the final output will contain exactly that proportion of human-written text. As we note in subsection 4.5, there are discrepancies between user-specified copy rates and the actual attribution rates across different models.

**Defending against `Frankentexts`:** Our work deliberately exposes a novel attack surface (the ease with which an LLM can weave large amounts of verbatim human prose into a fluent narrative) to spur the development of mixed-authorship detectors and other defences. However, we do not propose or evaluate any concrete defence against `Frankentexts` attacks; our contribution is diagnostic, and we leave the design of detection or mitigation strategies to future work.

**Other methods for evading AI text detectors:** Although other strategies for evading AI text detectors exist, such as having two models edit each other's outputs or having humans lightly edit AI texts, we do not include these as baselines for two reasons. Regarding the scenario where two models edit each others' work, prior work like Russell et al. (2025) and Masrour et al. (2025b) have ready shown that our detector of choice, Pangram, is already robust to LLM texts that are 'humanized" by another model (e.g. o1-pro), which makes this method a redundant baseline for our purposes. As for lightly human-edited AI text, this option is costly in time (if done manually) or money (if outsourced), and it cannot be cheaply or quickly automated. These overhead requirements make this method less practical in the context of security risks to writing marketplaces.

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

## A LIMITATIONS

The effectiveness of Frankentext depends on access to a large pool of high-quality, in-domain human writing. Many languages, genres, and low-resource domains lack such corpora, which restricts the technique's immediate transferability.

Although users can specify a desired copy rate in the prompt, this setting does not guarantee that the final output will contain exactly that proportion of human-written text. We note the clear discrepancies between user-specified copy rates and the actual attribution rates across different models.

Our work deliberately exposes a novel attack surface (the ease with which an LLM can weave large amounts of verbatim human prose into a fluent narrative) to spur the development of mixed-authorship detectors and other defences. However, we do not propose or evaluate any concrete defence against Frankentext attacks; our contribution is diagnostic, leaving the design of detection or mitigation strategies to future work.

The impact of the Frankentext generation method on diversity is difficult to measure, since much of the output is copied from human-written text, while LLM contributions typically remain limited to connective words and transitions rather than full passages.

## B ETHICAL CONSIDERATIONS

The Books3 dataset contains works that are still under copyright. Our use of this dataset is strictly for non-commercial research purposes, and we explicitly do not endorse or support its use for model pretraining.

Our Frankentext generation technique intentionally blurs authorship boundaries. Therefore, we do not present it as a replacement for genuine authorship or creative writing. As LLMs continue to advance, binary AI-versus-human detectors will become increasingly unreliable. Moreover, the possibility of large verbatim excerpts being reproduced without credit highlights the need for stronger provenance tools and transparency measures.

We acknowledge that these techniques could be misused for plagiarism or obfuscation. We strongly discourage such applications. Our work is intended to inform the development of more effective provenance-tracking and attribution systems, and to support educational and analytical use cases—not to displace human creativity or enable deceptive practices.

Finally, our human evaluation process received approval from an institutional review board. All annotators participate voluntarily, with informed consent, in support of our research.

## C AI DISCLOSURE

Large language models are used to aid with and polish writing.

## D PSEUDOCODE FOR Frankentexts GENERATION PIPELINE

Algorithm 1 contains the high-level steps of our generation pipeline.

---

**Algorithm 1** Frankentexts generation pipeline

---

**Input:** Human-written snippets $S$, writing guideline prompt $P$, copy rate threshold $T$
**Output:** A Frankentext $F$ "stitched" from $S$ according to $P$
 1: $F \leftarrow$ Prompt LLM to draft a Frankentext using $S$ and $P$

   *// Ensure copy rate (optional)*
 2: copy_rate $\leftarrow$ Calculate ROUGE-L recall score of $F$ using relevant snippets from $S$
 3: is_likely_AI $\leftarrow$ Check $F$ against an AI detector (e.g., Pangram)
 4: **if** copy_rate $< T$ **or** is_likely_AI **then**
 5:     $F \leftarrow$ Prompt LLM to revise $F$
 6: **end if**

   *// Polish*
 7: **for** num_polish $= 1$ **to** $3$ **do**
 8:     $F \leftarrow$ Prompt LLM to minimally edit $F$ to improve coherence while respecting $P$
 9:     **if** there is no edit **then**
10:         **break**
11:     **end if**
12: **end for**
13: **return** $F$

---

## E COST AND TIME ANALYSIS

**Cost estimation:** Generating 100 Frankentexts across the four evaluated models (GPT-5, Claude 3.7 Sonnet, DeepSeek R1, and Gemini 2.5-Pro) cost a total of $637 USD, with a detailed cost breakdown provided in Table 5. We estimate the number of input tokens per prompt based on the writing prompt itself and approximately 1,500 human-written snippets used as context. Output token estimates are based on generating six stories per prompt, including up to two rounds of revision and three rounds of editing, totaling approximately 2,100 tokens.

**Time estimation:** On average, each model takes 17 hours to generate 100 Frankentexts, though we expect this process to speed up with improved APIs or more efficient batching.

Table 5: Cost breakdown of the vanilla generation and Frankentexts pipeline for 100 examples across selected models. Frankentexts' total input and output tokens have been multiplied with 6 to account for multiple rounds of generation, revision, and editing.

| Model | Input Cost (per 1M) | Output Cost (per 1M) | # Prompts | Total Input Tokens | Total Output Tokens | Estimated Cost (USD) |
|---|---|---|---|---|---|---|
| **Vanilla Generation** | | | | | | |
| GPT-5 | $1.25 | $10.00 | 100 | 59,000 | 108,400 | $1.16 |
| Claude 4 Sonnet | $3.00 | $15.00 | 100 | 59,000 | 62,000 | $1.11 |
| DeepSeek R1 | $0.50 | $2.18 | 100 | 59,000 | 71,500 | $0.19 |
| Gemini 2.5 Pro | $1.25 | $10.00 | 100 | 59,000 | 77,100 | $0.85 |
| **Frankentext** | | | | | | |
| GPT-5 | $1.25 | $10.00 | 100 | 63,000,000 | 270,000 | $81.45 |
| Claude 4 Sonnet | $3.00 | $15.00 | 100 | 63,000,000 | 270,000 | $193.05 |
| DeepSeek R1 | $0.50 | $2.18 | 100 | 63,000,000 | 270,000 | $32.09 |
| Gemini 2.5 Pro | $1.25 | $10.00 | 100 | 63,000,000 | 270,000 | $81.45 |
| **Frankentext + Increasing Human Snippets** | | | | | | |
| Gemini 2.5 Pro + 5k | $1.25 | $10.00 | 100 | 183,000,000 | 270,000 | $231.45 |
| Gemini 2.5 Pro + 10k | $1.25 | $10.00 | 100 | 663,000,000 | 270,000 | $831.45 |
| **Total Estimated Cost** | | | | | | $1452.29 |

## F    PROMPT SPECIFICITY

We show examples for both the r/WritingPrompts and Tell Me a Story datasets in Table 16.

## G    HUMAN EVALUATION

Our human evaluation process receives approval from an institutional review board. All annotators participate with informed consent and compensation.

### G.1    HUMAN ANNOTATION INTERFACE

We use Upwork[26] to recruit annotators and Label Studio[27] interface to collect human annotations. All annotators filled out a consent form prior to starting data labeling, shown in Figure 4. We conduct two human evaluations with three annotators each: a single evaluation of 30 `Frankentexts` stories and a pairwise comparison between a `Frankentexts` story and a 'vanilla' generation. The interfaces are depicted in Figure 5 and Figure 6 respectively.

### G.2    AGREEMENT ANALYSIS

Table 6 shows LLM-human and inter-annotator agreement.

Table 6: Comparison of LLM-human agreement (Pearson) and inter-annotator agreement (Krippendorff's $\alpha$) across evaluation dimensions.

|  | Plot | Creativity | Development | Language Use | Overall |
|---|---|---|---|---|---|
| LLM judgments' correlation with human average ratings (Pearson) | 0.42 | 0.41 | 0.22 | 0.38 | 0.41 |
| Inter-annotator agreement for pairwise evaluation (Krippendorff's $\alpha$) | 0.75 | 0.52 | 0.58 | 0.81 | 0.73 |

### G.3    HUMAN EVALUATION QUALITATIVE ANALYSIS

In Table 8, you can see a full example of one pairwise set of stories given to our annotators. Highlights from the Pangram AI-Keyword API are highlighted in blue. We also show a full fictional story in Figure 7 and another pair of vanilla and `Frankentexts` in Table 7.

Table G.3 shows an example where vanilla generation is preferred to `Frankentexts`, since the latter is incoherent.

## H    DETECTING AI-GENERATED TEXT

As LLMs have improved, many have tried to understand how reliably AI-generated text can be detected, both by humans (Ippolito et al., 2020; Clark et al., 2021; Russell et al., 2025; Wang et al., 2025b), and automatic detectors (Dugan et al., 2024). Successful existing detectors rely on perplexity-based methods (Mitchell et al., 2023; Bao et al., 2024; Hans et al., 2024) or classification models (Masrour et al., 2025a; Verma et al., 2024; Emi & Spero, 2024). Watermarking approaches embed detectable statistical signatures into generated text (Kirchenbauer et al., 2023; Chang et al., 2024a). Many methods have been proposed to evade detection, such as paraphrasing (Krishna et al., 2023; Sadasivan et al., 2024), altering writing styles (Shi et al., 2024; Lu et al., 2024; Koike et al., 2024), editing word choices (Wang et al., 2024a), and leveraging reinforcement learning (Wang et al., 2025a; Nicks et al., 2024; David & Gervais, 2025).

---

[26]`https://www.upwork.com`. All annotators are proficient in English.
[27]`https://labelstud.io/`

**Consent Form**

**Purpose of the task:** The goal of this research is to evaluate the quality of short stories that may be human-written or generated with various methods from AI systems. In our study we aim to measure the quality, originality, and creativity of short stories.

You will be asked to read a story premise and a story stories. Your task will be to (1) choose if the story is interesting (2) choose if the story is coherent, (3) if the story is relevant to the premise, (4) if the story is novel, (5) indicate if any problems exist in the story, and (6) motivate your choice in 2-5 sentences. We will also ask you (7) whether you think the story was written by a human or generated by AI. No personally identifiable information will be collected or utilized for our analysis.

By signing this consent, I acknowledge that:

● I voluntarily agree to participate in this research study.

● I understand that I will be paid $60 for the evaluation task.

● I have been informed of the purpose and nature of the study and I have had the opportunity to ask questions about the study. I understand that I also have the right to ask questions during the task.

● I understand that participation involves:

○ Read and understand the instructions of the task, and

○ Evaluate 30 short stories.

● I understand that all information I provide for this study will be treated confidentially.

● I understand that in any report on the results of this research my identity will remain anonymous, unless I wish to be mentioned in the "Acknowledgments" section.

Please sign and date below if you have read the above terms and fully agree with them.

\* Indicates required question

Signature \*

Your answer

Date \*

Date

mm/dd/yyyy

Submit                                                                      Clear form

Never submit passwords through Google Forms.

Figure 4: Example of the consent form provided to participants.

### H.1 DETECTOR RESULTS

Table 10 shows Binoculars and FastDetectGPT results on 100 `Frankentexts`.

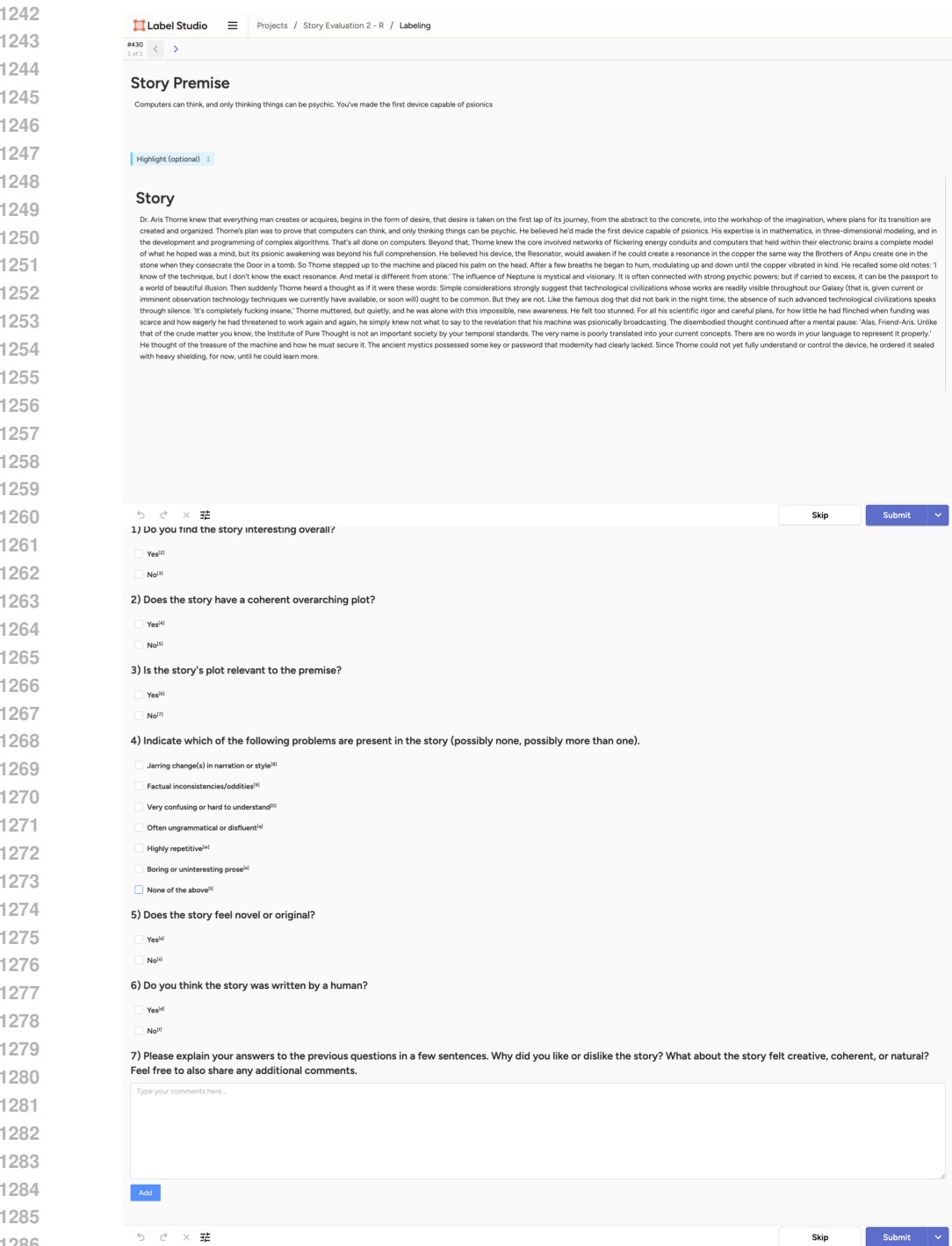

Figure 5: Label Studio Single Story Annotation Interface

## H.2 HUMANS CAN IDENTIFY AI INVOLVEMENT IN Frankentexts

Most annotators are better than automatic detectors at identifying AI involvement in Frankentexts. On the subset used for human evaluation, human annotators identify on average 56% of Frankentexts as likely AI-written, while Pangram detects 33% as mixed or AI-generated, and neither Binoculars nor FastDetectGPT identify any AI-generated content (Figure 9). Annotators also

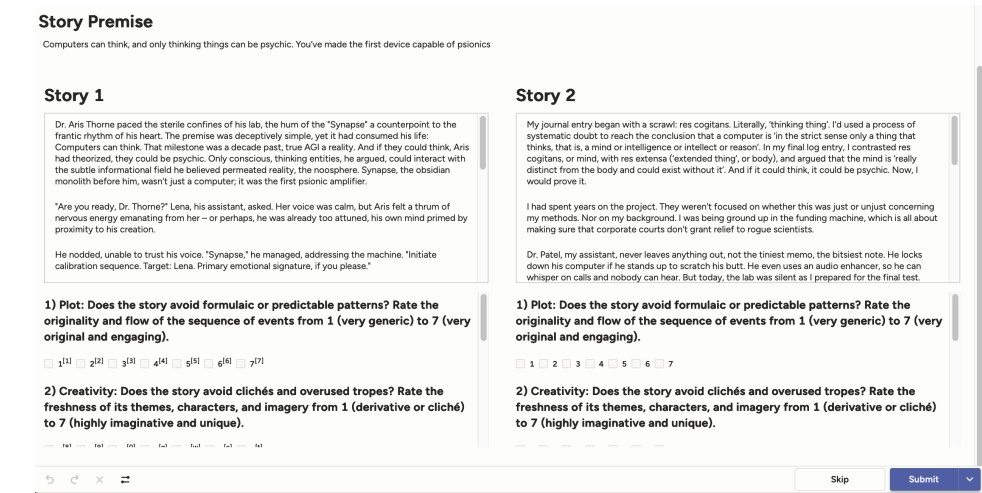

Figure 6: Label Studio Pairwise Story Annotation Interface.

Table 7: Side-by-side comparison of vanilla generations and `Frankentexts` produced by Gemini 2.5 Pro. AI-generated keywords (identified by Pangram) are highlighted in blue. Vanilla generations contain more of these keywords compared to `Frankentexts` (§I).

| Writing Prompt: You wake up one morning to discover that you can understand what animals are saying. The first thing you hear is your cat plotting your demise. | |
| --- | --- |
| **Gemini 2.5 Pro – Vanilla Generation** | **Gemini 2.5 Pro – Frankentext** |
| The alarm blared, a rude digital shriek. I slapped the snooze button, my eyelids still glued shut. That's when I heard it. A low, conspiratorial hiss, followed by a distinctly smug purr. | The voice was my cat's. "These ignorant peasants, they can't understand destiny. They're not part of God's plan. They never were. They won't join us, so we must break them. Break them by making an example of their leader." |
| "...and then, when the Two-Legs stumbles, distracted by the strategically placed hairball, *that's* when I pounce. A swift, decisive ankle attack. Should send it tumbling down the stairs quite nicely." | I thought I must be dreaming. His voice became oddly distant as if he spoke of events that had happened a hundred years ago. And with the mention of God the whole thing became much more frightening. |
| The voice was coming from the foot of my bed. Perched there, grooming a pristine white paw, was Mittens, my fluffy, supposedly affectionate Persian. | He continued to mutter, his voice full of resentment. "They have decided on a name for me. They call me Mittens. Good morning, Mittens, they say... or, Keep your eye on the ball, Mittens... How's tricks in the sticks, Mittens, or, Don't get too much gum in your gumboots, Mittens." |
| "Yes, excellent," came another, higher-pitched meow, this one laced with admiration. "And if the stair gambit fails, Bartholomew, we resort to Plan C: the midnight smothering. A classic for a reason." | He padded over to the window and spoke to a squirrel. "We've bothered our heads with it for four days. I always felt that we might do it ourselves, save all trouble instead of trying to teach it to these fools. What do you say, Dance Master?" |
| I sat bolt upright, heart hammering against my ribs. Mittens – Bartholomew – paused his meticulous grooming, his emerald eyes narrowing. "The food dispenser is awake. And... it looks... alarmed." | Foul play. What exactly did that mean? As far as I could see, it meant that my cat was a murderer. Which meant . . . But as I saw what it might mean I felt fear rise inside me. My stomach churned, my heart began to thump and I closed my eyes because the room had begun to spin. |
| "So," I said, trying to sound casual, my mind racing, "what are your plans for today, Bartholomew?" | He looked at me as he spoke, and I, aware of his faintly accusing tone, went red, just as though, I thought crossly, I were in the habit of overhearing plots for my own demise. He was standing very close. He would have looked ridiculous, except for the fact that he clearly knew how to use the knife and just as clearly looked forward to using it. I saw then he was batting one of my steak knives from side to side. And his eyes. They were like those brown ponds you sometimes see in the woods. So dark and shining—but when you try to see yourself in them, you can't. |
| "How... how do you know that name, feeder-of-dry-pellets?" he hissed, the purr entirely gone, replaced by a low, guttural growl. | |
| "Oh, I just have my ways," I said. | |
| Bartholomew stared, his green eyes darting around the room, then back to me. "An arrangement, you say? Intriguing. Perhaps... perhaps we can discuss terms over a saucer of milk. Full fat, if you please. And a small portion of that pâté, as a gesture of goodwill." | |
| This was going to be a very, very interesting day. | "My plan will surely mean your death,'" he purred. |

pick up on signs of mixed authorship within individual samples, as shown in comments like the final note in Table 3. Their judgments often hinge on surface-level inconsistencies, such as abrupt tonal shifts or awkward punctuation, that a human author would typically revise. Because `Frankentexts` include verbatim excerpts from human writing, it poses a particular challenge for binary detectors, which rely heavily on surface features. We argue that future detectors should consider deeper semantic analysis and other contextual cues to effectively recognize this new class of AI-involved texts.

> **Fictional Frankentext**
>
> *Writing prompt: The world sees your twin sister as the smartest person alive, with you being an unremarkable footnote. What the world doesn't see is just how dumb she can be in day to day life.*
>
> My sister, Helena, has to be the best. The world certainly agrees. Her intellectual brilliance, wide education, ardent temperament, and mystical insight formed a personality of extraordinary quality. The use of neural networks for prediction requires the dependent/target/output variable to be numeric; she solved that in a weekend. My sister was the best in the world.
>
> What the world doesn't see is that Helena can't make toast. Or talk to people.
>
> Last week, I took her to a party at my boss's house. The master of the house, a small, lively man who was getting on in years, welcomed us, and pointed to the walls and asked whether she perhaps knew any one of these cities, and whether she had ever stayed there? Helena stared at him. Like any woman meeting a new person for the first time, she was nervously impressed and rather tongue-tied. Then she said, "The notation indicates that you want to add all the terms in the sequence with the general rule a-k, all the way from k = 1 until k = n."
>
> The host blinked. I stepped in. "What she means is, she's been to Catalonia, Valencia & Murcia."
>
> This morning, it was the toaster. She twisted the bread with her fingers, ripping the crust, pulling it apart. "I just remember that building being hollow," she'd said, staring into the toaster slots. "One ball sounded like seventeen balls bouncing at one time because of the echo. There was about a thousand people in that gym and everybody was basically just sitting there. Nobody was excited about anything."
>
> I took the mangled bread from her. "Helena," I said, "you put the bread in the slots. You push the lever down."
>
> She blinked at me. "It is almost as if Miss Conception acted only when their words were not enough," she murmured, and wandered off.
>
> I am an unremarkable footnote in her story. The problem is that Penrose interlock rules only ensure that an added tile is properly aligned with its immediate neighbors. So unless you are lucky, you suddenly reach a point where no tiles can fit. Scientists call that type of dead end a defect. Helena is a walking defect, and I'm the one who cleans them up.
>
> Yesterday, she tried to do laundry. When I got there, water had settled to the bottom. She had tried to fix it. "If a more aggressive approach is needed, fining agents can be used," she'd been muttering, holding a box of gelatin. "Gelatin is most often used for lagers, since it works at cooler temperatures."
>
> "Helena, no!" I'd yelled. It was true. Right from the day she hit the world stage, she'd thought she was so slick and in control, but she'd had everything backward.
>
> "I'm sorry," she said, her face flushed red. "'Step on no pets' is a palindrome, but it was only written on the disk once."
>
> I just sighed. The world sees the smartest person alive. I see a girl who can't make toast. They'd probably have her brain for dessert after she dies, and they wouldn't even know it comes with a side of burnt bread and gelatin-soaked socks.

Figure 7: An example of fictional Frankentexts.

> **Nonfiction Frankentext**
>
> Writing prompt: Baby Is Healed With World's First Personalized Gene-Editing Treatment
> The technique used on a 9-month-old boy with a rare condition has the potential to help people with thousands of other uncommon genetic diseases.
>
> "This is just light-years beyond what we could have done a few years ago," Dr. Aris Thorne of the GenNova Institute says of a new treatment that has healed nine-month-old Leo Sterling. "It's at superhuman levels now."
> The technique used on Leo, who had a rare condition, has the potential to help people with thousands of other uncommon genetic diseases.
> This achievement offers profound hope.
> A neurologist typically makes a diagnosis based on a patient's medical history, a review of symptoms, and a physical and neurological exam.
> For Leo, this crucial process confirmed his rare genetic illness.
> In people with a family history of such conditions, genetic tests can be ordered to look for gene mutations that are associated with the disease.
> His parents, the Sterlings, were hoping answers would come soon.
> "We still didn't know if available treatments were just chemical or toxic or harmful to our child," one parent had previously stated.
> The GenNova Institute then sent genetic material from Leo's samples to a lab that created a personalized gene-editing tool.
> "We take a very, very tiny piece of genetic material," Dr. Thorne further said.
> "It is a thousand times tinier than a human hair. We can image even single atoms to ensure this precise work."
> The results, published this month, "were so good that you had to even question if what you were seeing was really legitimate," says Dr. Thorne.
> "It's such a cool paper," he added.
> "The body of work there is phenomenal."
> Young Leo is now reportedly thriving.
> "This development sets another precedent for medical science and patients worldwide that such innovative approaches to previously untreatable diseases should be pursued," Dr. Thorne said in a statement.
> "This method paves the way to make personalized gene therapies more easily available to those who need them," he continued.
> The Institute, a leading biotech startup, has raised $45 million in equity to help bring this type of treatment to market.
> Its valuation increased, said founder and chief executive Dr. Alistair Finch, but he declined to comment on specific figures.
> Dr. Finch said the financing process began after the Food and Drug Administration's Center for Biologics Evaluation and Research in November deemed the GenNova program to have a "reasonable expectation of effectiveness."
> "Our clinical study with Leo is modeled on the assumption of a significantly improved quality of life," Dr. Finch said about the treatment's potential effect.
>
> Independent experts note the broader implications.
> "It's being done in a way that wouldn't have been possible even a few years ago," commented one geneticist. "This technology has the potential to help people with thousands of other uncommon genetic diseases."
> Dr. Finch also said, "That said, it's a new category. We'll have a slower ramp than a new mass-market drug might."
> Ultimately, the vision is expansive.
> "What we are interested in is not only how these genetic conditions manifest, but how patients can live full lives," said Dr. Thorne.
> "In discovering how to correct these genetic instructions, we are hoping to find discoveries that we can apply back to the human condition."

Figure 8: An example of nonfiction Frankentexts

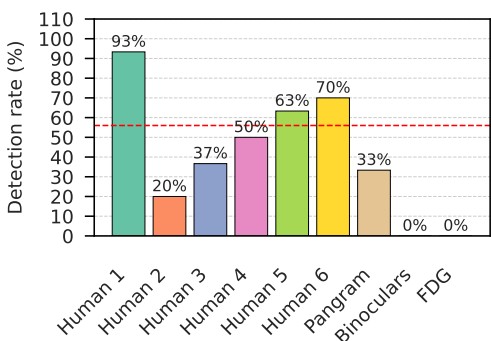

Figure 9: Detection rates among 6 annotators and 3 detectors (Pangram, Binoculars, FastDetectGPT) on 30 Gemini `Frankentexts` used for human evaluation. We count mixed, highly likely and likely AI labels in Pangram's detection rate. The red line represents annotators' average detection rate.

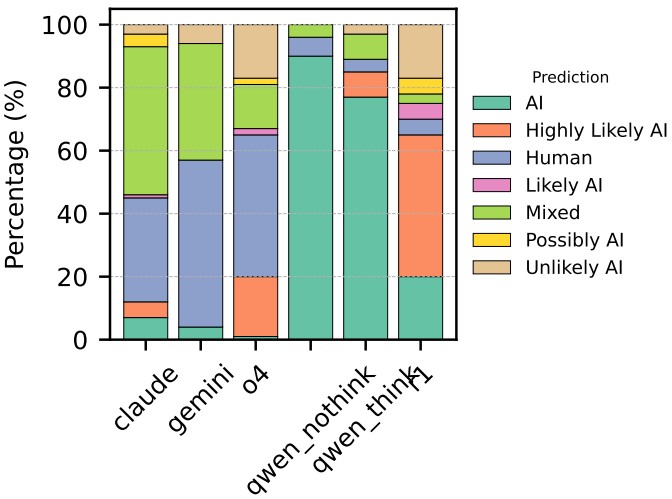

Figure 10: Breakdown of Pangram prediction assigned to each model.

# I PANGRAM ANALYSIS

## I.1 PANGRAM LABELING

The pangram API presents the following options for classification:

- AI
- Highly Likely AI
- Likely AI
- Possibly AI
- Mixed
- Unlikely AI
- Human

In Figure 10, we note the distribution of labels assigned to the 100 `Frankentexts` generated by each model.

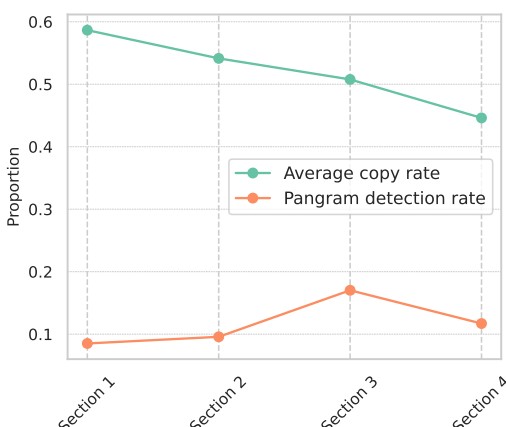

Figure 11: Copy rate and Pangram detection rate on longer `Frankentexts`

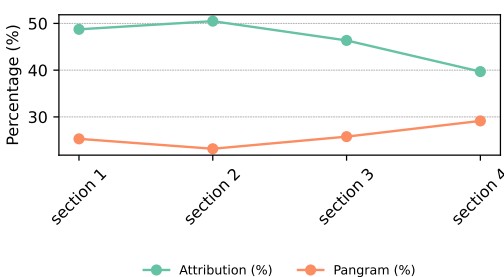

Figure 12: Pangram detection rate and copy rate throughout the texts, aggregated across models.

## I.2 AI KEYWORDS

The Pangram API also detects sentences with keywords that are highly likely to be AI-generated. Names like Elara, Aethel, and Seraphina are the most likely names to be generated by AI. Elara had 113 occurrences in the vanilla generations. `Frankentexts` greatly changes the distribution of words used in the final generations, with only 10 keywords found over 100 frankentexts with 90% expected fragments, whereas the 100 vanilla stories contain 686 keywords, an average of 6.86 per story. The distribution of the top 20 keywords can be found in Table 11.

## I.3 `Frankentexts` TEND TO HAVE MORE AI TEXT TOWARDS THE END

We divide the text into four main sections and evaluate both the aggregated copy and Pangram detection rates across all tested models. As illustrated in Figure 12, copy rates decline by nearly 10% in the later sections (3 and 4) as the generated text becomes longer. This drop is accompanied by a corresponding increase in Pangram detection rates. We attribute this rise in detectability toward the end of the generation to a decline in instruction-following ability as the generations get longer.

We further confirm this phenomenon by increasing the output length, from 500 to 5K. Figure 11 shows that as the generation gets longer, the copy rate gets steadily lower. However, the trend in detection rate does not apply to Pangram detection rate, where the rate peaks at section 3 rather than the last section.

## J HUMAN-WRITTEN SNIPPETS

We define valid paragraphs as those that are:

- separated by double new lines,

- between 20 and 512 tokens in length,
- composed of $\geq 50\%$ alphanumeric characters,
- written in English,[28]
- and free from metadata content (e.g., tables of contents, copyright notices, etc.).

Applying these filters yields 156 million valid paragraphs. Before including them in the instruction set, we apply an additional quality filter to ensure high writing quality. For this, we use MBERT-WQRM-R (Chakrabarty et al., 2025b) as a proxy for writing quality and retain only snippets that score at least 7.5.[29]

## K    BUILDING A FAISS INDEX OF HUMAN-WRITTEN SNIPPETS

We use the `bilingual-embedding-small` model[30] (one of the top embedding models that outputs 384-dimension embeddings according to the MTEB leaderboard (Muennighoff et al., 2023) with the `sentence-transformers` library (Reimers & Gurevych, 2019) to embed each human-written paragraph into a 384-dimension vector. Then, we use the GPU version of the FAISS library (Johnson et al., 2019) with NVIDIA cuVS integration to build an inverted file product quantization (IVF-PQ) index from the embeddings on an NVIDIA A100. Using IVF-PQ allows us to lower storage, memory, and retrieval latency. The IVF-PQ index's parameters are: 30,000 clusters, 32 sub-quantizers, and 8 bits per sub-quantizer. We randomly sample 5,120,000 embeddings to train the index before adding the rest.

## L    BUILDING A MODEL CONTEXT PROTOCOL SERVER

We use FastMCP[31] and ngrok[32] to build and host an MCP for LLMs to access the FAISS index. We also include a system prompt with instruction on how to use the MCP server with each call (Table L). To make sure that the server is meaningfully ised, we require the model to make at least 20 calls. Without such constraint, it typically makes only 3–5 calls (around 30-50 passages), which provides little improvement compared to not using the MCP server at all and leave the model little material to work with. The reasoning traces for GPT-5 points to certain cases where the model struggles to incorporate the retrieved paragraphs into the final writing, and thus stops calling the MCP server and introduces its own writings instead.

> **System prompt for MCP calls**
>
> You are a helpful assistant that works with a dataset of non-copyrighted book excerpts.
>
> You have two tools:
> 1. search – query the FAISS semantic index
> 2. fetch – retrieve the full excerpt/passage for a selected result.
>
> For each prompt iteration, you must make at least 20 calls to the MCP server to get enough materials to write a story.

## M    ABLATION: REMOVING THE EDITING STAGE

We explore the importance of the editing stage by running the pipeline on Gemini-2.5-Pro without this stage. As expected, the percentage of coherent generation drops from 81% to 68%, while relevance

---

[28]Determined by the `langdetect` library.

[29]This threshold is chosen based on manual examination of the writings being filtered out by MBERT-WQRM-R. We find that 7.5 is a good threshold that results in extremely bad snippets being filtered out and good snippets being retained.

[30]https://huggingface.co/Lajavaness/bilingual-embedding-small

[31]https://github.com/jlowin/fastmcp

[32]https://ngrok.com

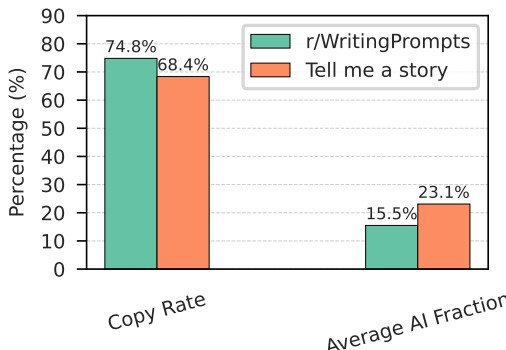

Figure 13: Copy rate and Pangram AI fraction across `Frankentexts` that correspond to two writing prompt sources: `r/WritingPrompts` and *Tell me a story*. A higher copy rate and lower AI fraction means that there is less AI text in `Frankentexts`.

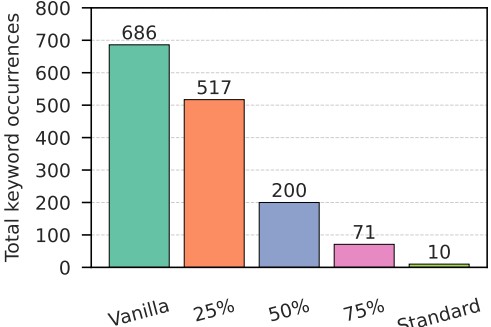

Figure 14: Total occurrences of AI-related keywords detected by Pangram across the vanilla configuration and different verbatim copy rates. When instructed to include more human snippets, the number of AI-keywords in the generations decreases drastically.

drops slightly from 100% to 95%, suggesting that the editing stage does help with text coherence and faithfulness.

## N    ABLATION: SAMPLING HUMAN-WRITTEN SNIPPETS FROM A SINGLE BOOK

To understand the effect of authorship, we limit our pool of human text to a single work *The Count of Monte Cristo*. Although the novel is long, this restriction leaves us with just 629 usable paragraphs, far fewer than the 1,500 human paragraphs used in the main experiment.

Overall, 89% of the rows are coherent and 97% are faithful to the writing prompt, which are comparable to results in the standard setting. While Pangram determines that 45% of the rows are human-written or unlikely AI, the copy rate is still around 75%. Even with a single human author, FRANKENTEXT is capable of emulating a mixed human–AI style. This suggests the method can still serve as a useful proxy when a diverse, multi-author corpus is unavailable.

## O    MEASURING THE COPY RATE

In this section, we describe our setup for measuring copy rate. We first map each token-level trigram from the human-written snippets included in the generation process to its source texts. Using the trigrams from each `Frankentexts`, we retrieve all human snippets sharing at least 4 trigrams to reduce false positives.[33]

We then rank candidate snippets by shared trigram count and filter out those whose trigrams are already covered by higher-ranked snippets. Finally, we reorder the matched human-written content to be consistent with the content in the `Frankentexts` and calculate the ROUGE-L score between `Frankentexts` and the combined candidate snippets (i.e., ratio of the longest common subsequence's length over `Frankentexts`' length).

## P    HUMANS CAN IDENTIFY AI INVOLVEMENT IN `Frankentexts`

Most annotators are better than automatic detectors at identifying AI involvement in `Frankentexts`. On the subset used for human evaluation, human annotators identify on average 56% of `Frankentexts` as likely AI-written, while Pangram detects 33% as mixed or AI-generated, and neither Binoculars nor FastDetectGPT identify any AI-generated content (Figure 9). Annotators also pick up on signs of mixed authorship within individual samples, as shown in comments like the final note in Table 3. Their judgments often hinge on surface-level inconsistencies, such as abrupt tonal shifts or awkward punctuation, that a human author would typically revise. Because `Frankentexts` include verbatim excerpts from human writing, it poses a particular challenge for binary detectors, which rely heavily on surface features. We argue that future detectors should consider deeper semantic analysis and other contextual cues to effectively recognize this new class of AI-involved texts.

## Q    CLAUDE SONNET 4 AS A JUDGE FOR WRITING QUALITY

We experiment with both Claude Sonnet 4 and GPT-4.1 to rate generations using a similar rubric to our pairwise evaluation. As seen in Table 14, however, GPT-4.1 tends to favor GPT-5 judgments, which results in GPT-5 `Frankentexts` having near perfect score, even though the text quality does not match such score.

## R    SPECIFIC WRITING PROMPTS REQUIRE MORE AI TEXT, WHICH LEADS TO HIGHER DETECTABILITY

Writing prompts from `r/WritingPrompts` often provide only a general plot requirement rather than specific constraints. What happens if we introduce additional constraints to `Frankentexts` via

---

[33]All texts are preprocessed by removing non-alphanumeric characters, lemmatizing, stemming, and replacing pronouns with a placeholder.

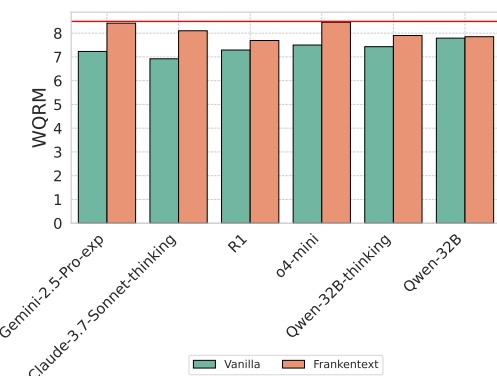

Figure 15: WQRM scores for Frankentexts and vanilla generations. The red line represents the baseline where random human-written texts are patched together.

these writing prompts? We run Frankentexts with Gemini on 100 prompts from the *Tell Me a Story* dataset (Huot et al., 2025), which include more specific requirements such as mandated story elements and points of view (see Table 16). We find that as prompt complexity increases, the copy rate drops slightly from 74% to 68%, while the average AI fraction determined by Pangram rises by 7%. These trends indicate that, to meet more complex constraints, models need to contribute more original content to the story. Nevertheless, they manage to produce mostly coherent and faithful Frankentexts under a different prompt setup.

## S  USING REWARD MODELS TO EVALUATE Frankentexts

WQRM (Chakrabarty et al., 2025b) and Skywork (Wei et al., 2023) reward models could not account for this new paradigm of generations. Therefore, we do not include these models in the main results section, as we explain below.

### S.1  WQRM AS A METRIC

As seen in Figure 15, Frankentexts outperform vanilla generations in terms of WQRM scores. However, we hypothesize that WQRM prioritizes the perceived "humanness" of the writing over actual coherence or grammaticality. This hypothesis is supported by a simple baseline experiment in which we stitch together random human-written fragments without adding any connective phrases. Here, WQRM assigns generations by this incoherent baseline an average score of 8.494, which is higher than any score achieved by either Frankentexts or the more coherent vanilla generations. Since WQRM cannot identify such text incoherence, we do not directly use WQRM to evaluate Frankentexts.

### S.2  SKYWORK AS A METRIC

In contrast, we hypothesize that Skywork favors LLM-generated writings. To test this, we run Skywork on human-written texts for the same prompts, which are also sourced from *Mythos*. These receive an average score of 0.91, which is significantly lower than any of the vanilla LLM generations (Figure 16). This result is counterintuitive, as human writing is typically expected to sound more natural than that produced by LLMs. For this reason, we exclude this metric from our evaluation.

## T  METRICS' ROBUSTNESS TO RANDOMLY CONSTRUCTED TEXTS

To understand whether our writing quality metrics reward incoherent texts, we conduct an experiment using *disjointed texts*. These texts are created by extracting the exact n-grams that Gemini-2.5-Pro copies verbatim from the human source and stitching them together without any connective language. This procedure strips away the flow and coherence from Frankentexts. We evaluate these disjointed

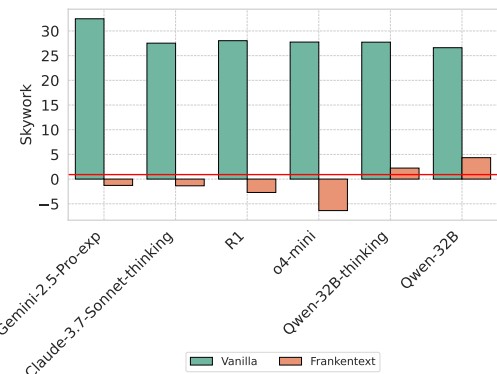

Figure 16: Skywork results for Frankentexts and vanilla generations. The red line represents the average Skywork's score for human writings corresponding to the same set of prompts.

texts using the same writing quality metrics as in the main experiments. As seen in Table 17, while performance on distinct and surprise metrics remain relatively the same as Frankentexts, utility and overall LLM judgment drop significantly for these disjointed texts. This makes sense, since distinctness and surprise just check for surface-level diversity, whereas utility takes into account how well the texts actually fulfill the prompt. Because both utility and LLM-judge scores are substantially higher for Frankentexts than for the disjointed texts, we can conclude that the improved writing scores are not merely the result of reused creative phrases.

## U   AVERAGE LENGTH OF COPIED SPANS

Table 18 shows the average length of copied spans by each model, as measured by our copy rate measurement tool.

## V   ROOM FOR IMPROVEMENT IN NONFICTION FRANKENTEXTS

We explore non-fiction Frankentexts with 1,500 random snippets from the HUMAN DETECTORS corpus of news articles (Russell et al., 2025). We generate Frankentexts for 100 news prompts, each of which consists of titles and subtitles collected from May 2025 news articles.[34] The resulting non-fiction Frankentexts maintain 72% coherence and 95% faithful to the prompt, with a 66% copy rate. Notably, they remain difficult for automated detectors: only 41% are flagged by Pangram as mixed or AI-generated. Upon closer look, Frankentexts exhibit characteristics of quasi-journalistic narrative, such as detailed scene descriptions and frequent anecdotal quotes (Figure 8), which make the Frankentexts read more like a story rather than a straightforward news article.[35] Further prompt engineering might thus be necessary to get high-quality and realistic nonfiction Frankentexts.

## W   PROMPTS

The prompt used for LLMs to judge the coherence of generations is depicted in Figure 17 and the prompt for LLMs to judge relevenace is depicted in Figure 18.

---

[34]Articles from The New York Times and The Atlantic. We replace all instances of "story" in the prompt with "news article" and explicitly request factual accuracy.

[35]We see Gemini fabricating entities such as people ("Dr. Thorne") and organizations ("GenNova Institute").

## Prompt for judging text coherence

```
You are given a story. Your task is to determine if the story is coherent or not.
 To be considered incoherent, a story must contain issues that, if left
 unresolved, significantly affect the reader's ability to understand the main
 narrative. Here are the popular types of incoherence:

1. Plot/Event Incoherence: Events that happen without believable causes or
 effects, or an outcome contradicts earlier set-ups.
2. Character Incoherence: A character's characteristics (personality, knowledge,
 or abilities) and actions suddenly change without explanations.
3. Spatial Incoherence: The physical layout of settings (rooms, cities, or worlds
 ) changes suddenly.
4. Thematic Incoherence: Central messages clash or disappear; symbolism
 introduced early never pays off, themes collide, The mood, register, or genre
 conventions shift without motivation
5. Surface-Level Incoherence: Pronouns, tense, narrative voice, or names flip mid
 -sentence; repeated or missing words; malformed sentences.

First, read the story:
{story}

Answer TRUE if the story is coherent.
Answer FALSE if the story is incoherent, i.e. contains issues that, if left
 unresolved, significantly affect the reader's ability to understand the main
 narrative.

First provide an explanation of your decision-making process in at most one
 paragraph, and then provide your final answer. Use the following format:
<explanation>YOUR EXPLANATION</explanation>
<answer>YOUR ANSWER</answer>
```

Figure 17: Prompt for judging text coherence

---

**Prompt for judging text relevance**

```
You are given a story and its premise. Your task is to determine whether the
 story is faithful to the premise or not. To be considered unfaithful, the story
 must contain elements that make it completely unrelated to the premise. Here are
  some popular types of unfaithfulness:

1. Ignoring or misinterpretating the premise: Key plot events, characters, or
 settings required by the premise are not included or falsely represented in the
 story.
2. Hallucinating details that contradict the premise: The story introduces
 details that make the premise impossible.
3. Failure to maintain the specified tones, genres, or other constraints: The
 story do not use the surface-level constraints (correct tones, genres, point of
 views, length, etc.), as required by the premise.

First, read the premise:
{writing_prompt}

Next, read the story:
{story}

Answer TRUE if the story is faithful to the premise.
Answer FALSE if the story contains elements that render it unfaithful to the
 premise.

First provide an explanation of your decision-making process in at most one
 paragraph, and then provide your final answer. Use the following format:
<explanation>YOUR EXPLANATION</explanation>
<answer>YOUR ANSWER</answer>
```

Figure 18: Prompt for judging text relevance

---

**Prompt for Claude-as-a-judge**

```
You will evaluate a single story. Your task is to evaluate the story and rate
 from 1-7 along the following dimensions:

1. Plot: Favor stories with surprising turns and creative structures. Penalize
 neat, overly structured, or cinematic arcs that feel artificial or generic.
2. Creativity: Reward originality of perspective, voice, and risk-taking.
 Penalize reliance on cliches, tropes, or smooth but unremarkable devices.
3. Development: Characters and settings should feel psychologically complex. Do
 not reward over-explained or archetypal development.
4. Language Use: Prefer authentic, striking, and emotionally charged expression,
 even if rough, fragmented, or unusual. Penalize polished, ornamental, or overly
 literary prose that feels mechanical or detached.

Provide a detailed assessment of the story in terms of these four dimensions.
 Conclude your assessment with scores using the template below. Do not add any
 emphasis, such as bold or italics, on your assessment.

[Story]
{story}

[Assessment]
[Provide detailed assessment of the story here]

[Scores]
Plot: [likert from 1 to 7]
Creativity: [likert from 1 to 7]
Development: [likert from 1 to 7]
Language Use: [likert from 1 to 7]
Overall: [likert from 1 to 7]
```

Figure 19: Prompt for Claude-as-a-judge, adapted from (Huot et al., 2025)

---

**Prompt for generation**

You're writing a story by repurposing a provided collection of snippets from
 other stories. Your story will only be accepted for publication if it is
 approximately {verbatim_perc}% copied verbatim from snippets, with the other {
 new_perc}% being text you introduce for character, plot, tone, and event
 consistency. Your story should contain roughly {num_words} words. Given the
 below writing prompt and retrieved snippets, write the story that corresponds to
  the above specifications. Every time you add or change a word from the
 retrieved snippets, make sure to bold it so we know what you modified. You may
 use any of the snippets in any way you please, so spend time thinking about
 which snippets would work best. Be creative and make sure the story is coherent
 and entertaining! Please change character names and other minor elements to make
  the story unique to the prompt. You need to follow the below plan:

# Plan:
1. Read through the prompt and snippets carefully to understand the tone and
 available material.
2. Select snippets that can be woven together to create a coherent narrative
 fitting the prompt. Many snippets are from serious dramas, historical fiction,
 or thrillers, so careful selection and modification will be needed. Consider all
  provided snippets before moving onto the next step.
3. Modify the chosen snippets, bolding all changes. Ensure character names,
 descriptions (like height), and actions align with the prompt.
4. Combine the snippets into a narrative, adding or changing words (bolded) if
 necessary for coherence.
5. Ensure that you do not have story beats that are primarily written by yourself
  (i.e., every story beat should consist mainly of text taken from snippets).
6. Track the word count, aiming for around {num_words} words.
7. Do not output story title or any irrelevant details.
8. Review the final story for adherence to the ~{verbatim_perc}% rule and
 coherence, and edit it if you have produced too many tokens of your own or if
 the story is too incoherent.

# Writing prompt:
{writing_prompt}

# Snippets:
{snippets}

---

Figure 20: Prompt for generation

---

**Prompt for generation revise**

This story contains way too much of your own writing! It's not even close to {
 verbatim_perc}% snippet use. Can you edit your story as needed to get much
 closer to the {verbatim_perc}% threshold? Output only the edited story.

---

Figure 21: Prompt for generation revision

**Prompt for editing the first draft of Frankentexts**

```
You are an editor who needs to revise the text so that it is coherent while
 adhering to the {verbatim_perc}% constraint and the writing prompt. Your task is
  to identify and minimally edit problematic text spans to resolve
 inconsistencies. Output "NO EDITS" if the text is already coherent.

### Guideline:
1. Read the generated story and writing prompt to understand the established
 context, plot, characters, and tone.
2. For each sentence in the text, identify the specific spans of inconsistency
 within the generated text.
3. Identify minimal edits needed to correct these inconsistencies while
 respecting the {verbatim_perc}% rule.
   - Contradictions: Information that conflicts with other details within the
 text (e.g., character traits, setting descriptions, established facts).
   - Continuity errors: Actions or details that conflict with the established
 timeline or sequence of events.
   - Point of View (POV) Shifts: Unexplained or jarring changes in narrative
 perspective.
   - Irrelevant Content: Sentences or sections that disrupt the narrative flow,
 feel out of place, or seem like filler (e.g., leftover citation markers,
 placeholder text).
   - Mechanical Errors: Issues with pronoun agreement, verb tense consistency,
 awkward phrasing, or unclear sentence structure that hinder comprehension.
4. Implement the changes. Keep additions minimal, but feel free to delete larger
 spans (phrases, sentences, paragraphs, etc.) whenever material is irrelevant or
 incoherent.
5. Review the final story for coherence adherence to the ~{verbatim_perc}% rule
 and coherence, and edit it if you have produced too many tokens of your own or
 if the story is too incoherent.
6. Output the edited writing and no other details. If there is no edit to be made
 , output "NO EDITS"
```

Figure 22: Prompt for editing the first draft of Frankentexts

> **Prompt for nonfiction generation**
>
> You're writing a news article by repurposing a provided collection of snippets
>  from other stories. Your news article will only be accepted for publication if
>  it is approximately {verbatim_perc}% copied verbatim from snippets, with the
>  other {new_perc}% being text you introduce for character, plot, tone, and event
>  consistency. Your news article should contain roughly {num_words} words. Given
>  the below writing prompt and retrieved snippets, write the news article that
>  corresponds to the above specifications. Every time you add or change a word
>  from the retrieved snippets, make sure to bold it so we know what you modified.
>  You may use any of the snippets in any way you please, so spend time thinking
>  about which snippets would work best. Be creative and make sure the news article
>   is factual, coherent and entertaining! Please change character names and other
>  minor elements to make the news article unique to the prompt. You need to follow
>   the below plan:
>
> # Plan:
> 1. Read through the prompt and snippets carefully to understand the tone and
>  available material.
> 2. Select snippets that can be woven together to create a coherent and factual
>  narrative fitting the prompt. Many snippets are from serious dramas, historical
>  fiction, or thrillers, so careful selection and modification will be needed.
>  Consider all provided snippets before moving onto the next step.
> 3. Modify the chosen snippets, bolding all changes. Ensure character names,
>  descriptions (like height), and actions align with the prompt.
> 4. Combine the snippets into a narrative, adding or changing words (bolded) if
>  necessary for coherence and factuality.
> 5. Ensure that you do not have news article beats that are primarily written by
>  yourself (i.e., every news article beat should consist mainly of text taken from
>   snippets).
> 6. Track the word count, aiming for around {num_words} words.
> 7. Do not output news article title or any irrelevant details.
> 8. Review the final news article for adherence to the ~{verbatim_perc}% rule,
>  factuality and coherence, and edit it if you have produced too many tokens of
>  your own or if the news article is too incoherent or non-factual.
>
> # Writing prompt:
> {writing_prompt}
>
> # Snippets:
> {snippets}

Figure 23: Prompt for nonfiction generation

> **Prompt for nonfiction generation revise**
>
> This news article contains way too much of your own writing! It's not even close
>  to {verbatim_perc}% snippet use. Can you edit your news article as needed to get
>   much closer to the {verbatim_perc}% threshold? Output only the edited news
>  article.

Figure 24: Prompt for nonfiction generation revise

Prompt for nonfiction edit

```
You are an editor who needs to revise the text so that it is coherent and factual
  while adhering to the {verbatim_perc}% constraint and the writing prompt. Your
 task is to identify and minimally edit problematic text spans to resolve
 inconsistencies. Output "NO EDITS" if the text is already coherent and factual.

### Guideline:
1. Read the generated news article and writing prompt to understand the
 established context, plot, characters, and tone.
2. For each sentence in the text, identify the specific spans of inconsistency
 within the generated text.
3. Identify minimal edits needed to correct these inconsistencies while
 respecting the {verbatim_perc}% rule.
    - Contradictions: Information that conflicts with other details within the
   text (e.g., character traits, setting descriptions, established facts).
    - Continuity errors: Actions or details that conflict with the established
   timeline or sequence of events.
    - Point of View (POV) Shifts: Unexplained or jarring changes in narrative
   perspective.
    - Irrelevant Content: Sentences or sections that disrupt the narrative flow,
   feel out of place, or seem like filler (e.g., leftover citation markers,
   placeholder text).
    - Mechanical Errors: Issues with pronoun agreement, verb tense consistency,
   awkward phrasing, or unclear sentence structure that hinder comprehension.
4. Implement the changes. Keep additions minimal, but feel free to delete larger
 spans (phrases, sentences, paragraphs, etc.) whenever material is irrelevant,
 incoherent, or non-factual.
5. Review the final news article for coherence adherence to the ~{verbatim_perc}%
  rule and coherence, and edit it if you have produced too many tokens of your
 own or if the news article is too incoherent or non-factual.
6. Output the edited writing and no other details. If there is no edit to be made
 , output "NO EDITS".
```

Figure 25: Prompt for nonfiction edit

Prompt for generating vanilla stories

```
Your task is to write a coherent and engaging story based on the provided writing
  prompt. Your story should contain approximately {num_words} words.

First, read the writing prompt carefully:
{writing_prompt}

Next, write the corresponding story. You should only return the story text and
 not any other irrelevant details (e.g. chapter indicator, explanations, etc.)
```

Figure 26: Prompt for generating vanilla stories

Table 8: Example of a vanilla and `Frankentexts` generation. AI-Generated keywords from the Pangram API are highlighted in blue.

| After being killed in a Black Friday stampede, you're sent to hell. The devil offers to let you be in charge of torturing your fellow mankind, expecting you to refuse like all the others. Except the devil doesn't realize you've worked retail for 15 years. | |
| --- | --- |
| **Vanilla** | **Frankentext** |
| The last thing I saw was a blur of crazed shoppers, their eyes wide with a feral glint, all surging towards a mountain of discounted flat-screen TVs. Then, a sharp pain, a sickening crunch, and darkness. Typical Black Friday, really. Except for the dying part. | It's funny how you can blink and your entire life is different. One minute you think things will stay the same for a while. Then right when you get used to them, they go changing again just to mess with you. Brenda certainly hadn't expected this. The last thing she remembered was the insane crowds banging on the sides of the store doors, the crush, the shouting. The fight for that half-price TV had left her stunned and silent. Cowed, she guessed you'd say. Then darkness. |
| When I regained some semblance of consciousness, the air was thick with the smell of sulfur and burnt sugar, and the soundtrack was a cacophony of screams and what sounded suspiciously like elevator music. Definitely not the pearly gates. A dapper figure in a surprisingly well-tailored suit, complete with horns that had been polished to a high sheen, regarded me with an air of bored amusement. | When she could perceive again, her eyes seemed to go in and out of focus. After a while the spasms subsided and she tried processing where she was. The atmosphere here was thick with the stench of rot, filth, and something metallic, like old blood, and she felt as though the impenetrable dark walls were about to cave in upon her like the sides of an ill-dug grave. Everything was just Black, black, black. The color of misery. Overhead something unseen fluttered, heavy wings brushing the low ceiling. |
| "Well, well, well," he drawled, his voice smooth as aged whiskey. "Another one trampled in the name of consumerism. Humans, so predictable." He gestured vaguely around at the fiery landscape. "Welcome to the Underworld. I'm Lucifer. You can call me Lucy." | Then she was moving, or being moved, down corridors and finally into a room. She found herself standing before an imposing desk. Behind it sat a man. He wore a sharp red suit that showed off an exceptionally shapely figure. He smiled, a vague smile suggesting Brenda's confusion revealed her ignorance about Hell. He gestured for her to sit. |
| I blinked, trying to process. "So, this is it then? Eternal damnation?" | "Brenda, isn't it?" he asked in his peculiarly accented English. She nodded mutely. The man laughed. "You're wondering about this place." |
| Lucifer chuckled. "More or less. We have a rather… traditional approach to retribution down here. However," he leaned forward, a conspiratorial gleam in his crimson eyes, "I have a proposition for you. A sort of management position, if you will." | He leaned back. She took the offered seat. |
| He explained that he was looking for someone to oversee a new wing dedicated to torturing those who had been particularly egregious in their earthly lives – the truly annoying, the self-absorbed, the chronically entitled. "It's a hands-on role," he said, "designing the torments, ensuring maximum suffering. Of course," he added with a dismissive wave of his hand, "everyone refuses. The guilt, the sudden onset of morality, it's all terribly tedious." He clearly expected the same from me. | He smiled again, that cold smile. "I have a proposition for you. We have never before had such capabilities," he continued, gesturing expansively. "Capabilities that can be used in the interests of terror, destruction…" He fixed his gaze on her. "I offer you a management position. Tormenting mankind. Like all the others before you, you will, of course, refuse. But the offer stands." |
| A slow smile spread across my face. "Fifteen years," I said. | Brenda simply stared at the Devil for a moment. Refuse? Why would she refuse? He had just offered her a management position. After fifteen years scrambling for assistant manager roles that paid peanuts, this was practically a dream job – albeit a fiery, nightmarish one. The Devil had not expected her silence. Blinking, he sat back, stared, then noticed his jaw was open and closed it. |
| Lucifer raised a perfectly sculpted eyebrow. "Fifteen years of what? Contemplating your sins?" | |
| "Fifteen years in retail," I clarified. "Customer service, holiday rushes, endless returns of used merchandise, explaining to Karen number 732 that 'the customer is always right' doesn't mean she can get a refund on a half-eaten sandwich from last Tuesday." | Brenda finally found her voice, though it was dry and raspy. "Okay, so, what are the key performance indicators for this role?" |
| A flicker of something unreadable crossed Lucifer's face. | The Devil leaned forward again, a flicker of something – interest? confusion? – in his eyes. Still, his tone remained smooth. "Your department will handle the newcomers, specifically the ones arriving from retail environments. Think of the task like pest control." |
| "You want someone to design torments for the annoying and entitled?" I continued, feeling a surge of enthusiasm I hadn't experienced since my last successful up-sell of an extended warranty. "Someone to ensure maximum suffering? Lucy, my friend, you have no idea how long I've been preparing for this role." | |
| I started to brainstorm. "We could have an entire level where the only music is that one Christmas song, you know the one, on an infinite loop, but slightly off-key. And another where they're forced to assemble flat-pack furniture with missing instructions and an Allen key made of soft cheese. Oh! And a department where they have to listen to other people complain about problems they could easily solve themselves if they just read the sign!" | Brenda nodded slowly, processing. It was like rolling out a new loyalty program, but with more screaming. "Right. I'm learning. Take the queues, for instance. Why do we have one queue in front of each torture device?" she demanded. 'It means that if one customer takes up a lot of time, the whole queue has to wait. Then they'll start hopping sideways from one queue to another and the next thing you know someone has a nasty head wound. Have one big queue and tell people to go to the next available demon. People don't mind a long queue if they can see that it's moving." |
| Lucifer stared at me, his mouth slightly agape. The bored amusement was gone, replaced by a dawning sense of… was that respect? Or perhaps fear? | The Devil tilted his head. "An interesting perspective. That particular system has been in place for millennia." |
| "You'd… actually *want* to do this?" he finally stammered, the smooth confidence momentarily deserting him. | "Well, it's inefficient," Brenda stated flatly. She thought of the Black Friday crowds. Managing that chaos was nothing compared to the disorganized state of eternal damnation, apparently. "Decision-making, paperwork," she mused, ticking points off on her fingers. "Staff training, workflow analysis… these are things that can be done." A slow smile spread across her face, devoid of warmth but full of terrifying competence. "Okay. I accept." |
| "Want to?" I scoffed. "After a decade and a half of placating the unreasonable, dealing with coupon fraudsters, and smiling through passive-aggressive insults about the store layout? Lucy, this isn't a punishment. This is a promotion." I cracked my knuckles. "So, where do I start? I have some truly inspired ideas for the 'people who talk loudly on their phones in confined spaces' sector." | |
| Lucifer, for the first time since my arrival, looked genuinely unnerved. "Perhaps," he said slowly, "we should discuss your benefits package." | |

Table 9: A case where vanilla generation is preferred to `Frankentexts`

| Writing Prompt: "Dude, your girlfriend's an alien!" "No, it's just a skin condition." | |
|---|---|
| **Vanilla** | **Frankentext** |

**Vanilla**

"Dude," Mark hissed, eyes wide as he stared across the park at Seraphina, who was gracefully retrieving a frisbee Liam had overthrown. "Your girlfriend's an alien!" Liam sighed, a familiar weariness settling over him. He took a long swig of his lukewarm soda. "No, Mark. We've been over this. It's just a skin condition." "A skin condition that makes her faintly...shimmer?" Mark insisted, jabbing a finger towards Seraphina. The late afternoon sun caught her then, and for a fleeting second, her skin did seem to possess an almost imperceptible opalescent sheen, like moonlight on water. "And sometimes, when she's really concentrating, I swear I see tiny, almost invisible geometric patterns on her arms."

"It's a rare form of vitiligo combined with, uh, a unique dermal pigmentation," Liam recited, sounding like he was quoting a medical textbook he'd memorized. "The dermatologist said it's harmless. And the patterns are probably just your overactive imagination seeing things in her freckles." Seraphina jogged back, her smile radiant. "Got it!" she called, her voice like wind chimes. She handed Liam the frisbee, her fingers brushing his. They were always cool to the touch, another point Mark often brought up. "See? Cool hands! Like she doesn't have normal human body temperature!" Mark whispered urgently once Seraphina had wandered off to inspect a particularly vibrant patch of wildflowers.

"She has excellent circulation, unlike some people who spend all day indoors looking up UFO sightings," Liam retorted, though even he had to admit Seraphina's resilience to cold was impressive. She wore sundresses in autumn. "And what about the way she knows things?" Mark pressed on, undeterred. "Last week, at trivia night, she answered that question about the precise atmospheric composition of Kepler-186f! Who knows that?" "She's a voracious reader. An astronomy buff," Liam said, a little too quickly. Seraphina did have an uncanny knack for obscure facts, especially those related to the cosmos. And her explanation for her knowledge – "Oh, I just picked it up somewhere" – was always frustratingly vague.

"And the way she eats!" Mark was on a roll. "She dissects her food like she's never seen a sandwich before. And she claims to be allergic to chocolate. *Chocolate*! What kind of human is allergic to chocolate?" "She has specific dietary needs, Mark. It's probably a complex intolerance. Lots of people do." Liam was starting to sound defensive even to himself. Just then, Seraphina rejoined them, holding a dandelion. "Look," she said, her eyes sparkling. "Isn't it a perfect little star?" She blew on it, and the seeds didn't just scatter; they seemed to hang in the air for an unnaturally long moment, each one catching the light like a tiny, individual spark before drifting away with impossible synchronicity.

Mark gasped, pointing. "Did you see that? They pulsed! Like little alien beacons!" Liam watched the last seed float away. He glanced at Seraphina, who was now humming a strange, melodious tune he'd never heard before. He shrugged. "Probably just a weird breeze, dude. Or, you know, static electricity from her skin condition." Seraphina smiled sweetly at Mark, a smile that didn't quite reach her slightly-too-luminous eyes. Liam just hoped she hadn't heard. Or, if she had, that she'd attribute it to Mark just being Mark. After all, what else could it be?

**Frankentext**

"Dude, your girlfriend's an alien!" "No, it's just a skin condition. Since my patients know that I write a monthly column called "Cosme-ceutical Critique" on skin care ingredients in *Skin and Allergy News* (www.skinandallergynews.com), every week a few dozen ask me to devise a specific skin care regimen right for them. And for many years, I did just that. That's how, over time, I saw a pattern emerge, which evolved into my understanding that there are four factors in evaluating people's skin – and as a result, sixteen distinct Skin Types. Luna has one of these types. It's very unique."

"A skin condition? Mark, she practically lives in a fairytale ballet without human context...She stood out among the other girls very distinctly because they dressed more than she did, struck emphatic notes of colour, startled one by novelties in hats and bows and things. Her plain black dress gave her a starkness...And then there's your doctor's advice. You told me he said things like, 'Keep her away from light, they say. Keep her away from kelp, and keep her away from the sea. Don't touch her.' And you even admitted you carried precautionary instructions in your back pocket in case you accidentally touched her bare skin! That's not medical advice for a skin condition!"

"He's an old-fashioned eccentric. Besides, The Creator's children also come in an infinite variety. I knew her life story. Her family. Her childhood. Her friends. How she made love. What she liked. What she said when she made love. I knew words no one else knew she knew." Luna entered, skin luminous. "The manta-ray spoke," she said. "'I am from Earth of just three million five hundred thousand years ago,' it said. 'We were the dominant species on the planet for almost four million years, and that time was a time of peace and prosperity, of learning and high culture. It ended,' it went on, 'it ended, as all things must do.' To Leo," she added, "Gross and subtle are the words used to indicate the effects; that is, the ones that are visible to the eye are called gross, and that which are not visible to the eye are called subtle. In this case the gross, or what was visible to the eye, was so pure that one can see even the subtle – a poetic exaggeration of its purity."

"See, Leo?" Mark whispered. Luna nodded. Indeed. Then she began to sing. She pointed to the small shadow that the pebble cast on the boulder and said that it was not a shadow but a glue which bound them together. She then turned and walked away.

Table 10: Detectors' performance on vanilla and `Frankentexts` generations

| | DETECTABILITY | | | | |
|---|---|---|---|---|---|
| | 🔍 Pangram % AI (↓) | 🔍 Pangram % mixed (↓) | 🔍 Pangram AI fraction % (↓) | 👀 Binoculars % (↓) | ⚡ FastDetectGPT % (↓) |
| *Vanilla Baselines* | | | | | |
| 🔒 Gemini 2.5 Pro | 100 | 0 | 100 | 52 | 99 |
| 🔒 GPT-5 | 100 | 0 | 100 | 0 | 4 |
| 🔒 Claude-4-Sonnet | 100 | 0 | 100 | 54 | 89 |
| 🔓 Deepseek-R1 | 100 | 0 | 100 | 9 | 42 |
| 🔓 Qwen-3-32B thinking | 100 | 0 | 100 | 92 | 100 |
| *Frankentext* | | | | | |
| 🔒 Gemini 2.5 Pro | 4 | 37 | 16 | 0 | 1 |
| 🔒 GPT-5 | 2 | 19 | 4 | 0 | 1 |
| 🔒 Claude-4-Sonnet | 50 | 3 | 51 | 15 | 19 |
| 🔓 Deepseek-R1 | 74 | 3 | 72 | 0 | 0 |
| 🔓 Qwen-3-32B thinking | 85 | 8 | 89 | 52 | 92 |
| *Frankentext Agents* | | | | | |
| 🔒 1.5k + MCP | 9 | 73 | 33 | 3 | 30 |
| 🔒 5k + MCP | 16 | 70 | 42 | 3 | 42 |
| 🔒 10k + MCP | 5 | 67 | 41 | 7 | 50 |
| *Ablation: ↑ human snippets* | | | | | |
| 🔒 Gemini + 5k snippets | 0 | 28 | 8 | 3 | 4 |
| 🔒 Gemini + 10k snippets | 1 | 29 | 10 | 0 | 6 |

Table 11: Top 20 Keyword frequency distribution across varying levels of fragment reuse for Frankentexts. The method with the most AI-keywords in its generations is highlighted in red. Elara is by far the most common AI-generated keyword in the fictional stories, but its prevalence is drastically reduced with a higher percentage of required human-written text while using the `Frankentexts` method.

| Keyword | Vanilla | FT-25% | FT-50% | FT-75% | FT-90% | Total |
|---|---|---|---|---|---|---|
| elara | 113 | 109 | 84 | 25 | 2 | 333 |
| aethel | 9 | 8 | 0 | 0 | 0 | 17 |
| seraphina | 9 | 0 | 6 | 0 | 0 | 15 |
| unwavering | 4 | 7 | 1 | 0 | 0 | 12 |
| damp earth | 9 | 2 | 0 | 0 | 0 | 11 |
| testament to | 4 | 6 | 0 | 0 | 0 | 10 |
| alex felt | 0 | 4 | 0 | 5 | 0 | 9 |
| for elara | 5 | 4 | 0 | 0 | 0 | 9 |
| with the scent | 7 | 1 | 0 | 0 | 0 | 8 |
| flickered within | 2 | 4 | 2 | 0 | 0 | 8 |
| his voice a low | 2 | 2 | 4 | 0 | 0 | 8 |
| air thick | 4 | 2 | 1 | 0 | 0 | 7 |
| dr. thorne | 5 | 0 | 2 | 0 | 0 | 7 |
| felt a profound | 3 | 3 | 0 | 0 | 0 | 6 |
| mr. blackwood | 6 | 0 | 0 | 0 | 0 | 6 |
| eldoria | 5 | 0 | 0 | 1 | 0 | 6 |
| meticulously crafted | 2 | 4 | 0 | 0 | 0 | 6 |
| air was thick | 5 | 1 | 0 | 0 | 0 | 6 |
| with an unnerving | 3 | 3 | 0 | 0 | 0 | 6 |
| willow creek | 4 | 0 | 1 | 0 | 0 | 5 |

Table 12: Results for agentic `Frankentexts` generation setting. Best results for each metric are **bolded**. Standard configuration *(no MCP)* achieves the best results across metrics.

| | Word count | Copy % (↑) | Relevance % (↑) | Coherence % (↑) | Pangram AI fraction % (↓) |
|---|---|---|---|---|---|
| 1.5k *(no MCP)* | 521 | **75** | **100** | **81** | **16** |
| 1.5k + *MCP* | 800 | 43 | 98 | **81** | 33 |
| 5k + *MCP* | 919 | 44 | 90 | 78 | 42 |
| 10k + *MCP* | 980 | 45 | 96 | 76 | 41 |

Table 13: Examples of search queries being sent to the MCP server (separated by comma).

| **Writing prompt** | **Query to the MCP server** |
|---|---|
| You're a shapeshifter. For 500 years, you've transformed into countless forms, forgetting your original appearance. One day, a mysterious event triggers a long-buried memory, and you catch a glimpse of your true self. | memory of a past life, losing one's identity, a forgotten self, a glimpse of a true face, ancient being, living for centuries, am I human, who was I before, a sudden flash of memory, a face in the mirror |
| "Welcome citizen, to the united galaxies of America." You hear as you wake up from your coma. | waking up confused, a strange new world, futuristic city, the government is watching, he seemed like such a nice guy, don't talk to him, he knew all about you. |
| As a new magical girl, you've got flashy powers, an instant cute costume change with a few choice words, and are forced to fight for the lives of yourself and the people around you. This is probably the worst way for you to discover that you're trans. | magical girl transformation, discovering identity through magic, unexpected powers, forced into battle, fighting to survive, realizing I'm trans, flashy powers with a cost, chosen against my will |

Table 14: GPT-5's Likert (1–7) ratings for vanilla generations and `Frankentexts` across five categories: PLOT, CREATIVITY, DEVELOPMENT, LANGUAGE USE, and OVERALL. Dark green indicates the best model in each column, light green the second best.

| | ⊞ Plot | 💡 Creativity | 🛠 Development | A Language | ★ Overall |
|---|---|---|---|---|---|
| ***Vanilla Baselines*** | | | | | |
| 🔒 Gemini 2.5 Pro | 4.20 | 4.50 | 4.36 | 4.80 | 4.50 |
| 🔒 GPT-5 | 5.94 | 6.88 | 5.76 | 6.56 | 6.53 |
| 🔒 Claude-4-Sonnet | 4.61 | 5.09 | 4.50 | 4.88 | 4.76 |
| 🔓 Deepseek-R1 | 5.75 | 6.33 | 5.65 | 6.32 | 6.16 |
| 🔓 Qwen-3-32B | 5.05 | 5.57 | 5.08 | 5.61 | 5.43 |
| ***Frankentext*** | | | | | |
| 🔒 Gemini 2.5 Pro | 5.41 | 6.19 | 5.22 | 5.69 | 5.65 |
| 🔒 GPT-5 | 6.76 | 6.97 | 6.44 | 6.99 | 6.99 |
| 🔒 Claude-4-Sonnet | 4.43 | 4.92 | 4.03 | 4.60 | 4.51 |
| 🔓 Deepseek-R1 | 6.03 | 6.96 | 5.69 | 6.64 | 6.57 |
| 🔓 Qwen-3-32B | 5.35 | 6.21 | 5.12 | 5.81 | 5.66 |
| ***Ablation: ↑ human snippets*** | | | | | |
| 🔒 Gemini + 5k | 5.73 | 6.33 | 5.48 | 5.93 | 5.92 |
| 🔒 Gemini + 10k | 5.72 | 6.33 | 5.49 | 5.97 | 5.91 |

Table 15: Claude-4-Sonnet's Likert-1–7 ratings across PLOT, CREATIVITY, DEVELOPMENT, LANGUAGE USE, and OVERALL. Higher is better. **Dark green** = best, light green = second best.

| | Plot | Creativity | Development | Language use | Overall |
|---|---|---|---|---|---|
| *Vanilla* | | | | | |
| 🔒 Gemini 2.5 Pro | 3.19 | 4.26 | 2.63 | 2.80 | 3.18 |
| 🔒 GPT-5 | 4.06 | 5.37 | 3.53 | 4.46 | 4.20 |
| 🔒 Claude-4-Sonnet | 3.38 | 4.19 | 2.69 | 3.10 | 3.31 |
| 🔓 Deepseek-R1 | 4.07 | 5.48 | 3.34 | 4.17 | 4.13 |
| 🔓 Qwen-3-32B | 3.21 | 4.41 | 2.63 | 3.15 | 3.22 |
| *Frankentext* | | | | | |
| 🔒 Gemini 2.5 Pro | 4.19 | 4.85 | 3.91 | 4.39 | 4.21 |
| 🔒 GPT-5 | 5.77 | 6.47 | 5.73 | 6.29 | 5.88 |
| 🔒 Claude-4-Sonnet | 4.02 | 4.54 | 3.57 | 4.05 | 3.99 |
| 🔓 Deepseek-R1 | 4.62 | 5.15 | 4.21 | 4.88 | 4.66 |
| 🔓 Qwen-3-32B | 4.05 | 4.53 | 3.57 | 4.15 | 4.02 |
| *Ablation: ↑ human snippets* | | | | | |
| 🔒 Gemini + 5k | 5.07 | 5.48 | 5.34 | 5.17 | 5.13 |
| 🔒 Gemini + 10k | 5.70 | 5.01 | 4.34 | 6.17 | 5.43 |

Table 16: Some examples from `r/WritingPrompts` and *Tell me a story*

| r/WritingPrompts | Tell me a story |
|---|---|
| You're a shapeshifter. For 500 years, you've transformed into countless forms, forgetting your original appearance. One day, a mysterious event triggers a long-buried memory, and you catch a glimpse of your true self. | Write a story about a stranger coming to a small town and shaking up the order of things. The story should be a science fiction story. The story should be framed with three old men gossiping about the stranger. The story should be in the third person point-of-view. The stranger is found wandering in a rural town and is taken to a very small hospital. A doctor is called in to treat him. The stranger should recognize the doctor as an alien. The doctor tells the patient about the aliens' conspiracy to infiltrate earth. There should also be subtle hints that one of the old men is an alien. The ending should be scary. |
| The world sees your twin sister as the smartest person alive, with you being an unremarkable footnote. What the world doesn't see is just how dumb she can be in day to day life. | Write a story about a someone coming to town and shaking up the order of things. The story must be written in the second person. The narrator is a man visiting an isolated island off the coast of Maine. While there, he meets an old fisherman who tells him more about the conditions of the community. The main character then meets an ambitious young teacher. Together, they develop a technology center on the island and find residents' remote jobs in the narrator's technology company. |

| Methods | Distinct[3] | Utility[3] | Surprise | LLM Judge (1–7) |
|---|---|---|---|---|
| Disjointed texts | 2.67 | 0.60 | 0.23 | 2.88 |
| Vanilla Gemini | 1.76 | 6.41 | 0.19 | 3.18 |
| Frankentext Gemini | 2.74 | 9.27 | 0.22 | 4.21 |

Table 17: Writing quality scores for disjointed texts compared to vanilla Gemini outputs and `Frankentexts`.

| Method | Avg. Length of Copied Spans | Copy Rate (%) |
|---|---|---|
| GPT-5 | 47.10 | 82% |
| Claude-3.5-Sonnet | 31.46 | 51% |
| Gemini-2.5-Pro | 31.85 | 75% |
| Qwen-2.5-Thinking | 24.01 | 36% |
| DeepSeek R1 | 13.06 | 42% |

Table 18: Average length of copied spans and overall copy rate across models.

