# OpenReview forum: "Frankentext: Stitching random text fragments into long-form narratives"
_ICLR.cc/2026/Conference — ICLR 2026 Conference Withdrawn Submission_

### Official Review · Reviewer_uMaE · 2025-10-27

**Soundness:** 2
**Presentation:** 4
**Contribution:** 3
**Rating:** 6
**Confidence:** 4

**Summary:**

This paper presents, Frankentext, a method of generating narrative that composes existing texts in response to a prompt instead of generating new text from scratch. Their system retrieves relevant passages from existing stories in response to a writing prompt or assembles sections from randomly chosen story excerpts with minimal transition phrases and editing added. The system is instructed what percentage of generated tokens should be copied verbatim from the existing texts. While this method reduces the coherence of the outputs, the stories are judged to be more diverse and original than typical LLM prompting. The resulting generations are also much harder for AI-generated text detectors to detect.

**Strengths:**

Originality:
- I haven't seen work like this before and I think it is an interesting task, and useful data for AI vs. human text detection

Quality:
- This paper presents a novel task clearly and enough experiments to demonstrate why this task is interesting (generations are more creative than vanilla LLM prompting).

Clarity:
- The paper is very well-written and clear.

Significance:
- This work will further research in AI text detection and understanding weaknesses in the creativity of AI text generations relative to humans.

**Weaknesses:**

1. From reading example outputs, they seem significantly less coherent than vanilla LLM generations. I think this could have been discussed or addressed more in the paper. Are the writing scores improving primarily based on including creative wording from the human-written text even though the overall outputs are not as coherent?

2. The ethical concerns with this work could be emphasized more. An obvious use of this work is to copy significant portions of existing work and fly under the radar of AI text detection systems. In particular, I would like to see statistics on the average length of copied span from existing text.

3. Intuitively, it seems MCP should improve over using random snippets. I am confused by the description of why this is not the case given in section 4.4. After retrieving relevant snippets, shouldn't the final task be formatted with the same prompt to the LLM as when there are random snippets? Why does the model suddenly become more verbose in this context?

4. Section 4.6 feels a bit like an afterthought since the issues describe may just be with prompting. I would consider cutting this section or moving the details to the appendix.

5. As much as possible, results discussed in the main body of the paper should be moved into the main body of the paper (e.g., Table 11).

**Questions:**

I appreciate that you specify in the ethical considerations that you do not endorse using this data for model pretraining based on copyright issues, but at another point in the paper you encourage the use of training AI vs. human text authorship detectors using this data. Given that these detectors could also be for commercial use, can you clarify how you see this use aligning with copyright concerns?

**Details Of Ethics Concerns:**

This work presents a methodology to generate stories by directly copying substantial portions of existing copyrighted works. The authors mention this in the body of their text and ethical considerations, but I am not familiar enough with copyright law to know if this is appropriate research to conduct with copyrighted material.

---

> ### Author Response · Authors · 2025-11-17
> **[1/1 -- Response to Reviewer uMaE] Evaluation, ethical discussion, and statistics on copied span length**
>
> We thank the reviewer for their thoughtful feedback. We are encouraged that the reviewer sees the work's originality and significance. Below, we address the reviewer's comments:
>
> **1. [Evaluation] Are the writing scores improving primarily based on including creative wording from the human-written text even though the overall outputs are not as coherent?**
>
> We include here an additional experiment to understand whether our writing quality metrics reward incoherent texts. Specifically, we run our metrics on disjointed texts, which are constructed by extracting the exact n-grams that Gemini-2.5-Pro copies verbatim from the human source and stitching them together without any connective language.
> | Methods             | Distinct³ | Utility³ | Surprise | LLM Judge (1–7) |
> |---------------------|-----------|----------|----------|------------------|
> | Disjointed texts    | 2.67      | 0.60     | 0.23     | 2.88             |
> | Vanilla Gemini      | 1.76      | 6.41     | 0.19     | 3.18             |
> | Frankentext Gemini  | 2.74      | 9.27     | 0.22     | 4.21             |
>
> While performance on distinct and surprise metrics remain relatively the same as Frankentexts, utility and overall LLM judgment drop significantly for these disjointed texts. This makes sense, since distinctness and surprise just check for surface-level diversity, whereas utility takes into account how well the texts actually fulfill the prompt. Because both utility and LLM-judge scores are substantially higher for Frankentexts than for the disjointed texts, we can conclude that the improved writing scores are not merely the result of reused creative phrases.
>
> **2. [Discussion] I would like to see statistics on the average length of copied span from existing text.**
>
> We thank the reviewer for their suggestion. We run our copy-rate metric on Frankentexts and report the average length of copied spans for each model. These values appear to closely follow the same trend as the copy rate. We will include this table and our Ethical Considerations section (Appendix B) in the main paper.
> | Method              | Average length of copied spans | Copy rate % |
> |---------------------|----------------------------------|-------------|
> | GPT-5               | 47.10                            | 82%         |
> | Claude-3.5-Sonnet   | 31.46                            | 51%         |
> | Gemini-2.5-Pro      | 31.85                            | 75%         |
> | Qwen-2.5-Thinking   | 24.01                            | 36%         |
> | DeepSeek R1         | 13.06                            | 42%         |
>
> **3. [Evaluation] After retrieving relevant snippets, shouldn't the final task be formatted with the same prompt to the LLM as when there are random snippets? Why does the model suddenly become more verbose in this context?**
>
> We are not completely sure what is causing this, mainly because we do not fully understand the synergy between the model's generation behavior and its tool-calling decisions. Our best guess is that, since the model is only making 3-5 MCP calls when we have constrained the prompt to make around 20 (see Section K), the model might be working with less information than it needs. Therefore, the model might have overcompensated by being extra wordy in the generations.
>
> **4. [Presentation] Section 4.6 can be moved to Appendix and Table 11 should be moved to the main paper.**
>
> We thank the reviewer for their suggestion and will revise the paper accordingly.
>
> **5. [Discussion] I appreciate that you specify in the ethical considerations that you do not endorse using this data for model pretraining based on copyright issues, but at another point in the paper you encourage the use of training AI vs. human text authorship detectors using this data. Given that these detectors could also be for commercial use, can you clarify how you see this use aligning with copyright concerns?**
>
> We understand the reviewer’s confusion. We emphasize that the dataset of random paragraphs will be provided directly by Frankentext users, so the users are free to use their own non-copyrighted data for creating Frankentexts for co-writing research or text detector training.
>
> Our use of Books3 in the paper serves solely to demonstrate how bad actors might exploit this technique to produce synthetic fiction capable of evading text-detection systems. We will revise the ethical considerations to explicitly state that we do not condone the use of copyrighted materials as inputs for producing Frankentexts.

---

> ### Author Response · Authors · 2025-11-26
> **Follow-up for Reviewer uMaE**
>
> Dear Reviewer uMaE,
>
> Thank you once again for your thoughtful review. If you have any follow-up questions or concerns regarding our rebuttal, we would be happy to provide additional clarification before the discussion period concludes.
>
> We have also **uploaded an updated version of the paper** (with highlighted changes), which incorporates your suggestions. To summarize, we have:
>
> - added Appendix T (line 1799-1811) to discuss the robustness of our writing quality metrics against disjointed texts.
> - added Appendix U (line 1812-1816) to report the average length of copied spans in Frankentext.
> - revised the Limitations and Ethical Considerations to clarify our stance against the use of copyrighted data for non-research purposes (line 544-551)
> - modified Appendix L to add more intuition on the agentic implementation’s results (line 1595-1599).
> - moved Section 4.6 to Appendix V and Table 11 to the main paper above Section 4.5.
>
> If our rebuttal has addressed your concerns, we would be grateful if you would consider adjusting your score accordingly. We sincerely appreciate your time and helpful feedback!

---

### Official Review · Reviewer_3dau · 2025-10-31

**Soundness:** 2
**Presentation:** 2
**Contribution:** 1
**Rating:** 2
**Confidence:** 4

**Summary:**

This paper introduces "Frankentexts," a novel paradigm for long-form narrative generation that reframes the role of a LLM from an author to a composer. The core task involves providing an LLM with a writing prompt and thousands of randomly sampled, human-written text fragments. The model is then constrained to produce a coherent narrative where a vast majority of the text (e.g., 90%) is copied verbatim from these fragments. The process involves the LLM implicitly searching the vast combinatorial space of snippets, creating an initial draft, and then iteratively "polishing" it for coherence while adhering to the copy constraint.

The authors conduct automatic and human evaluation on models such as Gemini 2.5 Pro, GPT-5, and Claude-4-Sonnet and they find that Frankentexts are often preferred by human ratings over unconstrained generation in terms of plot, creativity, and language use. The constrained text shows higher diversity, utility and surprise, but also lower relevance and especially coherence. They also conduct ablations, such as testing retrieval of relevant passages instead of adding thousands of random passages to the model, but this makes the model utilizing the retrieved passages less overall with no improvement in results. Finally, the authors note that this work poses a significant challenge to current AI detectors, since their composed texts cannot be detected as AI generated.

**Strengths:**

1. The authors have conducted thorough analysis on the benefits and shortcomings of their method via detailed human evaluation measuring different general-purpose and narrative-specific generation aspects. They also present several ablation studies to demonstrate the contribution of the different parts of their approach.
2. The paper poses an interesting question on AI text detection when LLMs are copying big portions from human written text.

**Weaknesses:**

1. The paper seems incomplete. The authors propose the idea of constraining LLM generation to mostly copying from human text (90% copying). First, this is not clearly motivated to me; what is the hypothesis when constraining generation in this way? More than that, the paper shows that this can produce interesting stories (but with lower coherence and grammatical issues, so there is a big trade off) only when conditioning on thousands of random passages. This is prohibitively expensive in order to be applied as a method for LLM generation, and when constraining the number of passages and applying retrieval of relevant documents instead which could be a more feasible solution, the authors show that this is not improving performance without any further insights.
2. Although the question of AI detectability in copied text is interesting, the authors do not really address this problem or investigate different detection methods. So, this aspect of the paper also is incomplete.

**Questions:**

1. What is the motivation/idea behind trying out this approach for generation? What is the hypothesis that you are trying to validate?
2. Have you done any analysis on why retrieving relevant passages is not helpful for the task? Does the LLM ignore the retrieved context (apart from not copying it over)?

---

> ### Author Response · Authors · 2025-11-17
> **[1/1 -- Response to Reviewer 3dau] Framing, scope, and retrieval error analysis**
>
> We thank the reviewer for their comments, and for acknowledging the thoroughness of our experiments as well as the interestingness of our topic. Below, we address the reviewer's concerns:
>
> **1. [Framing] What is the motivation/idea behind trying out this approach for generation? What is the hypothesis that you are trying to validate?**
>
> We began this work with a simple question: *Can we generate high-quality texts that can fool AI text detectors?* Our findings show that we can. Using Frankentexts, we have exposed vulnerabilities in current detection methods while presenting a non-traditional story generation approach that results in greater creativity and diversity than baseline models.
>
> Frankentexts’ balance between good writing quality and low detectability opens up applications like creating training data for AI text detectors and supporting co-writing research. Training data for text detectors is valuable because current detectors struggle with identifying which specific tokens are written by humans versus AI, and there is a shortage of datasets that provide such labels (something that Frankentexts can supply). In addition, Frankentexts offer a practical way to simulate human-AI co-written text, as discussed in Section 4.5 (lines 423-430). By adjusting the proportion of human to AI text during generation via the verbatim copy rate in the prompt, researchers can control the composition of the final outputs.
>
> Our results also highlight a technique that bad actors could use to produce narratives that can evade AI text detectors. In fact, AI-generated narratives have already flooded Amazon Kindle marketplace [[Knibbs-2024]](https://www.wired.com/story/scammy-ai-generated-books-flooding-amazon/), and the growing prevalence of such texts can threaten the livelihood of creative writing professionals [[Chakrabarty-2025]](https://ssrn.com/abstract=5606570)[[LitHub-2025]](https://lithub.com/against-ai-an-open-letter-from-writers-to-publishers/). As LLMs continue to improve, these risks will only intensify. We thus use this paper to call for stronger tools to track and regulate AI-generated content.
>
> We will revise the Introduction section (line 79-88) to make our contributions clearer.
>
> **2. [Scope] Although the question of AI detectability in copied text is interesting, the authors do not really address this problem or investigate different detection methods.**
>
> We have reported results for various state-of-the-art AI text detectors, including Pangram (Table 2) and Binoculars + FastDetectGPT (Table 8), which show that these detectors are vulnerable to Frankentexts. While it might be interesting to see how different detection methods do on Frankentexts, we focus on the detection methods that are most likely used by people, which reflect the real-world use case of Frankentexts.
>
> As noted in our Ethical Considerations section (Appendix B), our goal is to highlight the existence of such adversarial generation methods and to call for more effective attribution tools and legislation for LLM-generated content. Designing a detector that is robust to these texts is thus beyond the scope of this paper, and we leave it to future work to explore this direction.
>
> **3. [Experiments] Have you done any analysis on why retrieving relevant passages is not helpful for the task? Does the LLM ignore the retrieved context (apart from not copying it over)?**
>
> We attribute this to the fact that, despite being required to make at least 20 server calls, the model often produces only 3-5 calls, or around 30-50 passages (noted in Appendix K), which provides little improvement compared to not using the MCP server at all and leave the model little material to work with. Looking at the reasoning traces for GPT-5, we see that there are certain cases where the model struggles to incorporate the retrieved paragraphs into the final writing, and thus stops calling the MCP server and introduces its own writings instead.
>
> ---
> **References**
>
> [Knibbs-2024]
> Knibbs, Kate. “Scammy AI-Generated Book Rewrites Are Flooding Amazon.” WIRED, 10 Jan. 2024, www.wired.com/story/scammy-ai-generated-books-flooding-amazon/.
>
> [Chakrabarty-2025]
> Chakrabarty, Tuhin, Jane C. Ginsburg, and Paramveer Dhillon. Readers Prefer Outputs of AI Trained on Copyrighted Books over Expert Human Writers. Columbia Public Law Research Paper No. 5606570, 15 Oct. 2025. SSRN, https://ssrn.com/abstract=5606570.
>
> [LitHub-2025]
> Literary Hub. “Against AI: An Open Letter From Writers to Publishers.” Literary Hub, 27 June 2025, https://lithub.com/against-ai-an-open-letter-from-writers-to-publishers/.

---

> ### Author Response · Authors · 2025-11-26
> **Follow-up for Reviewer 3dau**
>
> Dear Reviewer 3dau,
>
> Thank you once again for your thoughtful review. If you have any follow-up questions or concerns regarding our rebuttal, we would be happy to provide additional clarification before the discussion period concludes.
>
> We have also **uploaded an updated version of the paper** (with highlighted changes), which incorporates your suggestions. To summarize, we have:
>
> - modified the Introduction section to clarify our motivation (line 41-46).
> - revised the Limitations and Ethical Considerations section to clarify that building defenses against Frankentexts is out of the scope for our paper (line 566-571).
> - modified Appendix L to add more intuition on the agentic implementation’s results (line 1595-1599).
>
> If our rebuttal has addressed your concerns, we would be grateful if you would consider adjusting your score accordingly. We sincerely appreciate your time and helpful feedback!

---

### Official Review · Reviewer_uxPN · 2025-10-31

**Soundness:** 3
**Presentation:** 3
**Contribution:** 3
**Rating:** 4
**Confidence:** 4

**Summary:**

This paper introduces Frankentext, a novel text generation pipeline where large language models (LLMs) assemble long-form narratives primarily by reusing human-written snippets verbatim. The proposed approach encourages models to stitch authentic paragraphs together with minimal connective phrases, aiming to produce outputs that appear more human-like and creative while evading AI-generated text detectors. Experiments with several advanced reasoning models (e.g., Gemini 2.5 Pro, GPT-5) demonstrate consistent improvements in perceived creativity, novelty, and textual quality compared to vanilla generation.

**Strengths:**

**1. Fascinating idea and topic:**

The work explores a very interesting paradigm, reusing human-authored texts as-is to enhance perceived authenticity and creativity.

**2. Strong empirical findings:**

The study shows that incorporating verbatim human text leads to outputs that evaluators and automatic metrics consider more creative and high-utility.

**3. Comprehensive evaluation:**

The experiments are extensive, involving multiple LLMs and consistent baseline comparisons.

**Weaknesses:**

**1. Limited topic relevance and coherence:**

The approach sometimes produces incoherent or context-mismatched passages. Since reward models such as WQRM or LLM-Judge can interpret any unusual text as “creative,” some outputs may be spuriously rewarded for randomness rather than genuine creativity.

**2. Complex and costly generation process:**

The method requires multiple inference rounds (i.e., drafting, revising, and polishing), which increases both computational cost and latency compared to ordinary text generation.

**3. Questionable definition of creativity:**

Although the resulting texts are evaluated as creative, they are not genuinely novel written materials (they primarily reassemble existing human). The framework therefore measures perceived creativity rather than true generative originality.

**4. Privacy and copyright concerns:**

Since the system directly retrieves and reproduces human texts from large corpora, it may expose copyrighted or private content. Ethical and legal implications should be discussed more deeply.

**Questions:**

See Weaknesses above.

---

> ### Author Response · Authors · 2025-11-17
> **[1/1 -- Response to Reviewer uxPN] Creativity definition & evaluation, methodology, and discussion**
>
> We appreciate the reviewer’s thoughtful comments and their recognition of the interest and breadth of our work. Below, we address their comments:
>
> **1. [Evaluation] Some outputs might be spuriously rewarded for randomness rather than genuine creativity by reward models such as WQRM and LLM judges.**
>
> We appreciate the reviewer raising this point and will include a discussion in the next version of the paper. In our study, we show that both human annotators and LLMs judge our texts as creative, not just the models themselves. In our study, LLM judgments show a moderate positive correlation with human ratings (Pearson r = 0.41). Human annotators rate the Gemini Frankentexts at an average of 5.10 for creativity on a 1-7 scale (Figure 2), while LLM judges rate the texts at an average of 4.85 (Table 13).
>
> We also run an experiment to understand whether random outputs will be spuriously rewarded by our metrics. To do this, we run the metrics on disjointed texts that are constructed by extracting the exact n-grams that Gemini-2.5-Pro copies verbatim from the human source and stitching them together without any connective language.
> | Methods             | Distinct³ | Utility³ | Surprise | LLM Judge (1–7) |
> |---------------------|-----------|----------|----------|------------------|
> | Disjointed texts    | 2.67      | 0.60     | 0.23     | 2.88             |
> | Vanilla Gemini      | 1.76      | 6.41     | 0.19     | 3.18             |
> | Frankentext Gemini  | 2.74      | 9.27     | 0.22     | 4.21             |
>
> While performance on distinct and surprise metrics remain relatively the same as Frankentexts, utility and overall quality drop significantly for these disjointed texts. This makes sense, since distinctness and surprise just check for surface-level diversity, whereas utility takes into account how well the texts actually fulfill the prompt. Therefore, we believe that both utility metric and LLM judges are robust to randomly constructed or incoherent inputs.
>
> **2. [Methodology] The method requires multiple inference rounds (i.e., drafting, revising, and polishing), which increases both computational cost and latency compared to ordinary text generation.**
>
> Frankentexts indeed cost more than vanilla generations to generate, but we consider these additional expenses a reasonable investment in producing high-quality data for our proposed applications, including training text detectors and doing co-writing research. Even in a potential misuse context, a bad actor could justify this cost to obtain high-quality texts at scale, especially given that each Frankentext costs only around one US dollar to generate. Furthermore, inference costs for frontier models are continuously declining, making such applications more financially feasible in the future.
>
> **3. [Evaluation] Although the resulting texts are evaluated as creative, they are not genuinely novel written materials (they primarily reassemble existing human). The framework therefore measures perceived creativity rather than true generative originality.**
>
> [[Boden-2004]](https://doi.org/10.4324/9780203508527) argues that there are three forms of creativity: combinatorial, exploratory, and transformational. Combinatorial creativity involves combining existing concepts or ideas; exploratory creativity concerns navigating within an established conceptual space; and transformational creativity requires reshaping that space itself.
>
> The reviewer appears to emphasize exploratory creativity, whereas Frankentexts lean more on the combinatorial creativity. However, both of them are still forms of creativity at the end of the day. We argue that identifying, selecting, and rearranging the right source material requires a meaningful degree of generative creativity, and thus cannot be dismissed as mere mechanical work.
>
> **4. [Discussion] Since the system directly retrieves and reproduces human texts from large corpora, it may expose copyrighted or private content. Ethical and legal implications should be discussed more deeply.**
>
> We appreciate the reviewer’s suggestion. In our approach, users have full control over the corpora they provide to the LLM, including the choice of whether to include copyrighted or private materials. Our use of Books3 in the paper serves solely to demonstrate how malicious actors might exploit this technique to produce synthetic fiction capable of evading text-detection systems. We will revise the ethical considerations to explicitly state that we do not condone the use of copyrighted or private content as inputs for producing Frankentexts.
>
> ---------
> **References**
>
> [Boden-2004]
> Boden, Margaret A. The creative mind: Myths and mechanisms. Routledge, 2004. https://doi.org/10.4324/9780203508527

---

> ### Author Response · Authors · 2025-11-26
> **Follow-up for Reviewer uxPN**
>
> Dear Reviewer uxPN,
>
> Thank you once again for your thoughtful review. If you have any follow-up questions or concerns regarding our rebuttal, we would be happy to provide additional clarification before the discussion period concludes.
>
> We have also **uploaded an updated version of the paper** (with highlighted changes), which incorporates your suggestions. To summarize, we have:
>
> - added Appendix T (line 1799-1811) to discuss the robustness of our writing quality metrics against disjointed texts.
> - revised our Limitations and Ethical Considerations section to discuss the cost implications of the method (line 553-558) as well as the use of copyrighted and private datasets (line 544-550).
>
> If our rebuttal has addressed your concerns, we would be grateful if you would consider adjusting your score accordingly. We sincerely appreciate your time and helpful feedback!

---

### Official Review · Reviewer_kEfY · 2025-11-02

**Soundness:** 3
**Presentation:** 2
**Contribution:** 3
**Rating:** 4
**Confidence:** 4

**Summary:**

This paper introduces FrankenTexts, texts that are compiled by a LLM from several different human-authored texts. The authors show through human and automated evaluations that the resulting texts are in many respects better than vanilla LLM-authored texts with the same writing prompt and that AI detectors identify them as human-authored.

**Strengths:**

The paper is well-written and contains many interesting experiments. The human and automated evals are thorough and the figures and diagrams are well-designed.

**Weaknesses:**

This paper is very interesting, but feels like a FrankenText of multiple different papers. The paper can't seem to make up its mind if it's about a generation methodology for fiction or a critique of AI detection methodologies. Under both interpretations, key experiments are missing, and I feel like the paper would be significantly stronger if it focused on one or the other.

Under the interpretation that the primary interest is in using this to write fiction, the paper does provide evidence that Frankentexts are preferred to vanilla texts, but that's not surprising to me (contra the authors' intuition in line 55). LLMs are generally regarded as worse writers than good humans, and by asking the model to copy as much as possible verbatim from published books the authors ensure a significantly higher writing quality than what a model would be able to do on their own. While I am glad that the authors compare to a vanilla baseline, it seems highly relevant to compare to the baseline of retrieval augmented generation where the model retrieves the same texts but isn't instructed to copy them verbatim. I'm not an expert in LLM-powered fiction writing, but whatever the current state-of-the-art in that area should be compared to as well under this interpretation. Furthermore, regardless of what one thinks of the ethical status of LLM generations in general, using this methodology to generate fiction is unambiguously plagiarism. The authors don't seem to acknowledge this fact anywhere.

Under the interpretation that the primary interest is in critiquing AI detectors, the authors fail to compare with the retrieval baseline again, as well as light human editing or two models editing each others' work. It's also unclear whether this really works as a critique of AI detectors at all given that 90%+ of the text is genuinely human-authored. At no point do the authors engage with the question of whether it is correct to view these FrankenTexts as AI-authors or human-authored. They also fail to engage in discussions related to the reprecusions and implications of the work that I would expect of a critique of AI detectors. Finally, although AI-detectors have had success marketing themselves as commercial tools, I do not know any AI researchers who seriously believe that they are reliable in the face of adversarial generations.

**Questions:**

1. Do you recommend this methodology as a way to generate fictional texts? If so, what do you say to the contention that this is plagiarism?
2. If 90% of a text is human-authored, why do you think it should be considered an AI-authored text?
3. Is there a meaningful difference in the distributions of texts that result from asking a human to provide the same task with the same sources?
4. What do these results mean, at the end of the day? What should I take from this paper off into my life? It feels like this paper is "just asking questions" when it should be answering them.

---

> ### Author Response · Authors · 2025-11-17
> **[3/3 -- Response to Reviewer kEfY] References**
>
> Here are the references that we use in our response to Reviewer kEfY:
>
> [Knibbs-2024]
> Knibbs, Kate. “Scammy AI-Generated Book Rewrites Are Flooding Amazon.” WIRED, 10 Jan. 2024, www.wired.com/story/scammy-ai-generated-books-flooding-amazon/.
>
> [Chakrabarty-2025]
> Chakrabarty, Tuhin, Jane C. Ginsburg, and Paramveer Dhillon. Readers Prefer Outputs of AI Trained on Copyrighted Books over Expert Human Writers. Columbia Public Law Research Paper No. 5606570, 15 Oct. 2025. SSRN, https://ssrn.com/abstract=5606570.
>
> [LitHub-2025]
> Literary Hub. “Against AI: An Open Letter From Writers to Publishers.” Literary Hub, 27 June 2025, lithub.com/against-ai-an-open-letter-from-writers-to-publishers/.
>
> [Huot-ICLR25]
> Huot, Fantine, Reinald Kim Amplayo, Jennimaria Palomaki, Alice Shoshana Jakobovits, Elizabeth Clark, and Mirella Lapata. “Agents’ Room: Narrative Generation through Multi-step Collaboration.” ICLR 2025 Conference, 22 Jan. 2025, OpenReview, https://openreview.net/forum?id=HfWcFs7XLR.
>
> [Chakrabarty-CHI24]
> Chakrabarty, Tuhin, Philippe Laban, Divyansh Agarwal, Smaranda Muresan, and Chien-Sheng Wu. “Art or Artifice? Large Language Models and the False Promise of Creativity.” Proceedings of the 2024 CHI Conference on Human Factors in Computing Systems, Association for Computing Machinery, 2024, pp. 1–34, https://doi.org/10.1145/3613904.3642731.
>
> [Paech-2024]
> Paech, Samuel J. “EQ-Bench: Emotional Intelligence Benchmark for LLMs.” EQ-Bench, https://eqbench.com/about.html.
>
> [Russell-ACL25]
> Russell, Jenna, Marzena Karpinska, and Mohit Iyyer. “People Who Frequently Use ChatGPT for Writing Tasks Are Accurate and Robust Detectors of AI-Generated Text.” Proceedings of the 63rd Annual Meeting of the Association for Computational Linguistics (Volume 1: Long Papers), Association for Computational Linguistics, July 2025, pp. 5342–5373. https://doi.org/10.18653/v1/2025.acl-long.267.
>
> [Jabarian-Imas-NBER25]
> Jabarian, Brian, and Alex Imas. Artificial Writing and Automated Detection. NBER Working Paper 34223, Sept. 2025, National Bureau of Economic Research, https://www.nber.org/papers/w34223.
>
> [Russell-arXiv25]
> Russell, Jenna; Marzena Karpinska; Destiny Akinode; Katherine Thai; Bradley Emi; Max Spero; and Mohit Iyyer. “AI Use in American Newspapers Is Widespread, Uneven, and Rarely Disclosed.” arXiv, Oct. 2025, https://arxiv.org/abs/2510.18774.
>
> [Masrour-GenAIDetect25]
> Masrour, Elyas, Bradley N. Emi, and Max Spero. “DAMAGE: Detecting Adversarially Modified AI Generated Text.” Proceedings of the 1st Workshop on GenAI Content Detection (GenAIDetect), Jan. 2025, Abu Dhabi, UAE, International Conference on Computational Linguistics, pp. 120–133. https://aclanthology.org/2025.genaidetect-1.9/.
>
> [Dugan-arXiv24]
> Dugan, Liam, et al. “RAID: A Shared Benchmark for Robust Evaluation of Machine-Generated Text Detectors.” arXiv, 10 June 2024, https://doi.org/10.48550/arXiv.2405.07940.
>
> [Ahn-NBER25]
> Ahn, David S., et al. Generative AI and the Future of Work: Evidence from the Introduction of ChatGPT. NBER Working Paper No. 34255, National Bureau of Economic Research, 2025, www.nber.org/system/files/working_papers/w34255/w34255.pdf.
>
> [Clio-Arxiv24]
> Tamkin, Alex, Miles McCain, Kunal Handa, Esin Durmus, Liane Lovitt, Ankur Rathi, Saffron Huang, Alfred Mountfield, Jerry Hong, Stuart Ritchie, Michael Stern, Brian Clarke, Landon Goldberg, Theodore R. Sumers, Jared Mueller, William McEachen, Wes Mitchell, Shan Carter, Jack Clark, Jared Kaplan, and Deep Ganguli. “Clio: Privacy-Preserving Insights into Real-World AI Use.” arXiv preprint, arXiv:2412.13678, 18 Dec 2024, https://arxiv.org/abs/2412.13678.

---

> ### Author Response · Authors · 2025-11-17
> **[2/3 -- Response to Reviewer kEfY] Experiments**
>
> **5. [Experiments] Baseline of retrieval augmented generation where the model retrieves the same texts but isn't instructed to copy them verbatim.**
>
> We thank the reviewer for their insightful suggestion and agree that this is a great baseline. We implemented it and provide results for this additional baseline below, implemented using Gemini-2.5-Pro. We include in each prompt 1,500 relevant paragraphs retrieved from the FAISS index originally constructed for our MCP implementation (see Appendix J, lines 1468-1478). The generation and editing prompts have been modified to remove the verbatim copying constraint.
>
> | Method | Word Count | Copy % | Relevance % | Coherence % | Distinct³ | Utility³ | Surprise | LLM Judge (1-7) | Pangram % Human |
> |-------|------------|--------|-------------|-------------|-----------|----------|----------|------------------|------------------|
> | Vanilla Baseline | 593 | — | 100 | 100 | 1.76 | 6.41 | 0.19 | 3.18 | 0 |
> | RAG (no verbatim copying) | 538 | 0.63 | 100 | 99 | 1.56 | 6.43 | 0.20 | 3.46 | 2 |
> | Frankentext | 521 | 75 | 100 | 81 | 2.74 | 9.27 | 0.22 | 4.21 | 59 |
>
> In terms of writing quality, the RAG baseline performs comparably to the vanilla baseline, but underperforms Frankentexts. This approach also offers no benefit in terms of detection, as Pangram misclassifies only two of its generations as human-written. These results suggest that the RAG baseline does not contribute meaningfully to our goal of producing high-quality texts that can fool text detectors. We will discuss these results in the next revision of the paper.
>
> **6. [Experiments] Whatever the current state-of-the-art in fiction generation should be compared to as well.**
>
> We follow the standard practice in modern narrative-generation research by comparing our generation method to stories produced by frontier LLMs, which is consistent with prior work on story-generation systems [[Huot-ICLR25]](https://openreview.net/forum?id=HfWcFs7XLR)[[Chakrabarty-CHI24]](https://doi.org/10.1145/3613904.3642731). Our choice also reflects real-world usage, as these models are the primary tools that most people rely on to produce large volumes of AI-generated text for public platforms.
>
> Our baselines also represent some of the strongest methods for this task. Benchmarks such as EQBench [[Paech-2024]](https://eqbench.com/about.html), which is possibly the most visible leaderboard in creative writing, show that the models we evaluate are among the top performers for this task. This is expected, as major labs invest significant money and effort to improve writing capabilities, given that “Writing” is among the most popular use cases for mainstream LLM chatbots like ChatGPT [[Ahn-NBER25]](https://www.nber.org/papers/w34255) and Claude [[Clio-Arxiv24]](https://arxiv.org/abs/2412.13678).
>
> **7. [Experiments] The authors fail to compare with light human editing or two models editing each others' work.**
>
> We thank the reviewer for their suggestion. Regarding the baseline where two models edit each others' work, prior work [[Russell-ACL25]](https://doi.org/10.18653/v1/2025.acl-long.267)[[Masrour-GenAIDetect25]](https://aclanthology.org/2025.genaidetect-1.9/) has already shown that Pangram (our evaluation method) is robust to LLM texts "humanized" using o1-pro. As for lightly edited human baselines, this option is costly in time (if done manually) or money (if outsourced), and it cannot be cheaply or quickly automated. As we noted above, this makes it less practical in the context of security risks to writing marketplaces.
>
> **8. [Experiments] Although AI-detectors have had success marketing themselves as commercial tools, I do not know any AI researchers who seriously believe that they are reliable in the face of adversarial generations.**
>
> We understand the reviewer’s concern, as many commercial AI detectors are indeed unreliable under adversarial conditions. Our use of Pangram, however, is based on a body of peer-reviewed research. Prior work, such as [[Russell-ACL25]](https://doi.org/10.18653/v1/2025.acl-long.267)[[Jabarian-Imas-NBER25]](https://www.nber.org/papers/w34223)[[Russell-arXiv25]](https://arxiv.org/abs/2510.18774)[[Dugan-arXiv24]](https://doi.org/10.48550/arXiv.2405.07940), shows that Pangram is more robust than typical detectors, including against strong “humanization” strategies such as o1-pro. These studies show that Pangram is reliable across model families, genres, and adversarial settings, which makes it a meaningful and appropriate tool for our evaluation.

---

> > ### Comment · Reviewer_kEfY · 2025-11-27
> > **Response 2/3**
> >
> > **5. [Experiments] Baseline of retrieval augmented generation where the model retrieves the same texts but isn't instructed to copy them verbatim.**
> >
> > Thanks for running this experiment! I agree with your interpretation of the results and am excited to see it included in the paper.
> >
> > **6. [Experiments] Whatever the current state-of-the-art in fiction generation should be compared to as well.**
> >
> > > We follow the standard practice in modern narrative-generation research by comparing our generation method to stories produced by frontier LLMs, which is consistent with prior work on story-generation systems [Huot-ICLR25][Chakrabarty-CHI24]. Our choice also reflects real-world usage, as these models are the primary tools that most people rely on to produce large volumes of AI-generated text for public platforms.
> > >
> > > Our baselines also represent some of the strongest methods for this task. Benchmarks such as EQBench [Paech-2024], which is possibly the most visible leaderboard in creative writing, show that the models we evaluate are among the top performers for this task. This is expected, as major labs invest significant money and effort to improve writing capabilities, given that “Writing” is among the most popular use cases for mainstream LLM chatbots like ChatGPT [Ahn-NBER25] and Claude [Clio-Arxiv24].
> >
> > I don't think you understand my criticism here. I'm saying that, if the primary goal of the paper is to propose a way to write fiction, you should show how this method compares the existing state-of-the-art. Huot et al. (2025) is actually an excellent paper to compare your method to. Without comparisons like this, I can't tell if this is a genuine advance or if it is just something better than the naive approach. Appealing to common usage doesn't countermand this criticism.
> >
> > **That said, since you've clarified that the primary goal is examining AI detectors and not generating novel fiction this is less important.** In this context comparing to how people typically use models makes a lot of sense. I still think that comparisons with other computational methodologies would be interesting, but it's not required.
> >
> > **7. [Experiments] The authors fail to compare with light human editing or two models editing each others' work.**
> >
> > > We thank the reviewer for their suggestion. Regarding the baseline where two models edit each others' work, prior work [Russell-ACL25][Masrour-GenAIDetect25] has already shown that Pangram (our evaluation method) is robust to LLM texts "humanized" using o1-pro. As for lightly edited human baselines, this option is costly in time (if done manually) or money (if outsourced), and it cannot be cheaply or quickly automated. As we noted above, this makes it less practical in the context of security risks to writing marketplaces.
> >
> > After reviewing the cited articles I agree that the suggested experiment isn't needed. Please include this information in the paper.
> >
> > **8. [Experiments] Although AI-detectors have had success marketing themselves as commercial tools, I do not know any AI researchers who seriously believe that they are reliable in the face of adversarial generations.**
> >
> > > We understand the reviewer’s concern, as many commercial AI detectors are indeed unreliable under adversarial conditions. Our use of Pangram, however, is based on a body of peer-reviewed research. Prior work, such as [Russell-ACL25][Jabarian-Imas-NBER25][Russell-arXiv25][Dugan-arXiv24], shows that Pangram is more robust than typical detectors, including against strong “humanization” strategies such as o1-pro. These studies show that Pangram is reliable across model families, genres, and adversarial settings, which makes it a meaningful and appropriate tool for our evaluation.
> >
> > After reviewing the linked papers I agree that Pangram is very good and withdraw this criticism.

---

> ### Author Response · Authors · 2025-11-17
> **[1/3 -- Response to Reviewer kEfY] Framing, discussion, and evaluation**
>
> We thank the reviewer for their detailed comments. We are encouraged that the reviewer found the paper to be well-written with interesting and thorough experiments. Below, we address the reviewer's concerns:
>
> **1. [Framing] The paper can't seem to make up its mind if it's about a generation methodology for fiction or a critique of AI detection methodologies.**
>
> We understand the reviewer’s confusion and will clarify this further in the next version of our paper. In short, our paper is motivated by a simple question: *Can we generate high-quality texts that can fool AI text detectors?* Our findings show that we can. Using Frankentexts, we have exposed vulnerabilities in current detection methods while presenting a non-traditional story generation approach that results in greater creativity and diversity than baseline models.
>
> Frankentexts’ balance between good writing quality and low detectability opens up applications like creating training data for AI text detectors and co-writing research. More importantly, our results highlight a technique that bad actors could use to produce narratives that can evade AI text detectors. In fact, AI-generated narratives have already flooded Amazon Kindle marketplace [[Knibbs-2024]](https://www.wired.com/story/scammy-ai-generated-books-flooding-amazon/), and the growing prevalence of such texts can threaten the livelihood of creative writing professionals [[Chakrabarty-2025]](https://ssrn.com/abstract=5606570)[[LitHub-2025]](https://lithub.com/against-ai-an-open-letter-from-writers-to-publishers/). As LLMs continue to improve, these risks will only intensify. We thus use this paper to call for stronger tools to track and regulate AI-generated content.
>
> We will revise the Introduction section (line 79-88) to make our contributions clearer.
>
> **2. [Discussion] At no point do the authors engage with the question of whether it is correct to view these FrankenTexts as AI-authors or human-authored.**
>
> We appreciate the reviewer’s comment. While we briefly discuss authorship in the Introduction (lines 83-88), we will discuss the nuances more in depth in the next version of the paper. In short, we do not think that there is a definitive answer about authorship, since different contexts can result in different interpretations. If authorship is defined by the amount of human effort involved, Frankentexts should be considered AI-generated, since all humans do is prompt the model. This perspective is particularly relevant when considering potential market harm to human authors, especially since such texts can be produced at scale with minimal human effort. However, if authorship is defined by whether most of the output originated from human-written text, one could argue they are largely human-written.
>
> **3. [Discussion] The authors have not discussed ethical implications for plagiarism or for AI detectors.**
>
> We address these issues in the Ethical Considerations section on page 18 of Appendix B, which we will move into the main text. To summarize, we acknowledge that Frankentexts and similar techniques could be misused for plagiarism or to obscure authorship. Our goal, however, is to call for stronger provenance-tracking and attribution tools, not to offer a substitute for authentic authorship or creative writing.
>
> **4. [Evaluation] The paper does provide evidence that Frankentexts are preferred to vanilla texts, but that's not surprising to me.**
>
> We understand the source of confusion. However, we emphasize that it is not obvious Frankentexts should be preferable to vanilla texts, given the restrictive and highly unnatural constraints under which the former are created. Composing a coherent collage from verbatim texts written in different contexts is an extremely difficult task - both for smaller, non-reasoning models (as noted in Footnote 7) and for humans - due to the sheer number of random input paragraphs. On average, each Gemini Frankentext is stitched together from 11 different sources, using verbatim chunks of about 32 tokens each. We will revise the Introduction (line 45-48) to further emphasize the difficulty of the task.

---

> ### Author Response · Authors · 2025-11-26
> **Follow-up for Reviewer kEfY**
>
> Dear Reviewer kEfY,
>
> Thank you once again for your thoughtful review. If you have any follow-up questions or concerns regarding our rebuttal, we would be happy to provide additional clarification before the discussion period concludes.
>
> We have also **uploaded an updated version of the paper** (with highlighted changes), which incorporates your suggestions. To summarize, we have:
> - modified the Introduction section to clarify our motivation (line 41-46) and emphasize our task difficulty (line 52-60).
> - moved the Limitation & Ethical considerations section to the main paper (line 528-571) to discuss the nuances of authorship and possible misuse of Frankentexts, including plagiarism.
> - modified the main experiment table (Table 2) to include results on RAG baselines.
> - added footnote 8 (line 263-266) to discuss our choice of creative writing baselines
> - modified Section 3.3 (line 277-279) to cite relevant work showing the robustness of Pangram (our detectability metric).
>
> If our rebuttal has addressed your concerns, we would be grateful if you would consider adjusting your score accordingly. We sincerely appreciate your time and helpful feedback!

---

> ### Comment · Reviewer_kEfY · 2025-11-27
> **Response 1/3**
>
> **1. [Framing] The paper can't seem to make up its mind if it's about a generation methodology for fiction or a critique of AI detection methodologies.**
>
> The edits do make this clearer, but my preference would be for a more comprehensive rewrite. I understand that that's unreasonable to expect during rebuttals and am more speaking to what I think the best version of this paper would look like.
>
> **2. [Discussion] At no point do the authors engage with the question of whether it is correct to view these FrankenTexts as AI-authors or human-authored.**
>
> > We appreciate the reviewer’s comment. While we briefly discuss authorship in the Introduction (lines 83-88), we will discuss the nuances more in depth in the next version of the paper. In short, we do not think that there is a definitive answer about authorship, since different contexts can result in different interpretations. If authorship is defined by the amount of human effort involved, Frankentexts should be considered AI-generated, since all humans do is prompt the model. This perspective is particularly relevant when considering potential market harm to human authors, especially since such texts can be produced at scale with minimal human effort. However, if authorship is defined by whether most of the output originated from human-written text, one could argue they are largely human-written.
>
> This response exemplifies what I am saying. You can't refuse to engage with the core ideas underpinning the paper and jsut say "well it depends on your POV." **What do you think?** Make an argument and convince me you're right. Cite prior writings on this topic. Engage in the philosophical conversation. This isn't ancillary to the point of your paper -  the answer to this question fundamentally shapes how one should view your work. Your paper is implicitly assuming an answer to this question. Is that an answer you're not willing to defend? Can you even tell me what the implicit assumption is?
>
> You seem particularly worried about LLMs writing books, but don't publishers have checks for if text is verbatim copied from previously existing books? How strong are those, and can they detect this? Can I steal a paragraph from Great Expectations and put it in a book of mine and publish it? The answer to this question is very important to whether this approach has any actual utility in the primary context you're pitching it in.
>
>
> **3. [Discussion] The authors have not discussed ethical implications for plagiarism or for AI detectors.**
>
> > We address these issues in the Ethical Considerations section on page 18 of Appendix B, which we will move into the main text. To summarize, we acknowledge that Frankentexts and similar techniques could be misused for plagiarism or to obscure authorship. Our goal, however, is to call for stronger provenance-tracking and attribution tools, not to offer a substitute for authentic authorship or creative writing.
>
> This work has implications for people using plagiarism / AI-authorship detectors. That's what I'm asking you to discuss, and it's not covered in either your appendix or the updated main text.
>
> **4. [Evaluation] The paper does provide evidence that Frankentexts are preferred to vanilla texts, but that's not surprising to me.**
>
> > We understand the source of confusion. However, we emphasize that it is not obvious Frankentexts should be preferable to vanilla texts, given the restrictive and highly unnatural constraints under which the former are created. Composing a coherent collage from verbatim texts written in different contexts is an extremely difficult task - both for smaller, non-reasoning models (as noted in Footnote 7) and for humans - due to the sheer number of random input paragraphs. On average, each Gemini Frankentext is stitched together from 11 different sources, using verbatim chunks of about 32 tokens each. We will revise the Introduction (line 45-48) to further emphasize the difficulty of the task.
>
> I'm not confused, I'm telling you that this result is obvious and intuitive to me. Footnote 7 says that they struggle to reliably follow the instruction. My claim is that a text assembled in this fashion should be obviously preferred to a vanilla generation. These are not the same claim. If you wish to provide evidence against my intuition you could show that, among texts that do follow the instructions generated by a weaker model, those are not preferred to vanilla generations.

---

> > ### Author Response · Authors · 2025-11-29
> > **[Response 1/3 first follow-up -- Reviewer kEfY] Framing, discussion, and evaluation**
> >
> > **1. [Framing] The paper can't seem to make up its mind if it's about a generation methodology for fiction or a critique of AI detection methodologies.**
> >
> > > The edits do make this clearer, but my preference would be for a more comprehensive rewrite. I understand that that's unreasonable to expect during rebuttals and am more speaking to what I think the best version of this paper would look like.
> >
> > We have further edited the Abstract, Introduction  and Methodology section to clarify our framing of Frankentexts as high-quality narratives that could evade AI text detectors.
> >
> > **2. [Discussion] At no point do the authors engage with the question of whether it is correct to view these FrankenTexts as AI-authors or human-authored.**
> >
> > > This response exemplifies what I am saying. You can't refuse to engage with the core ideas underpinning the paper and jsut say "well it depends on your POV." What do you think? Make an argument and convince me you're right....
> >
> > We hereby take a clear position: Frankentexts fall into the hybrid category of mixed human-AI writings, rather than neatly into either “AI-generated” or “human-written” extremes. This classification is grounded in their method of construction, rather than a fine-grained analysis of stylistic/semantic features of the final texts. Prior work has also already treated hybrid or AI-assisted texts as a distinct category, rather than forcing them into a binary “AI vs human” framing [[Wang-IJCAI25](https://arxiv.org/abs/2403.03506)][[Saha-ACL25](https://arxiv.org/abs/2502.15666)]. Other examples of texts in the same hybrid category include texts that are co-written by LLMs and humans, where LLMs are used for brainstorming and editing [[Chatterji-NBER25](https://www.nber.org/papers/w34255)], and AI texts that have been humanized by another model or edited by humans, as we already discussed earlier.
> >
> > Frankentexts should not be treated as purely human writing, as AI’s original work is still credited (though not to the same extent as human’s contribution) in similar scenarios where humans supply the writing and LLMs do the editing [[He-CHI25](https://dl.acm.org/doi/full/10.1145/3706598.3713522)]. Frankentexts are also not purely AI generation, as they contain a significant amount of human writings, as shown by our copy rate metric (roughly 75% of the tokens in Gemini Frankentexts can be traced back to the human-authored paragraphs).
> >
> > This dual nature of Frankentexts raises concerns along both dimensions of authorship. On the one hand, since an LLM orchestrates most of Frankentexts’ assembly, they can saturate the self-publishing platforms and thus harm the likelihood of creative writing professionals with original works. On the other hand, since Frankentexts contain a substantial proportion of human texts, they can also trigger copyright and plagiarism concerns.
> >
> > We have edited the first paragraph of the Limitations and Ethical Considerations section (“Authorship”) to discuss this viewpoint.
> >
> > > You seem particularly worried about LLMs writing books, but don't publishers have checks for if text is verbatim copied from previously existing books?
> >
> > When discussing harms from LLM-generated books, we are mainly concerned about self-publishing platforms (rather than traditional publishers), such as Amazon’s Kindle Marketplace/Kindle Direct Publishing and Archive of Our Own (AO3). These platforms have expressed concerns about AI-generated content, and Amazon has even explored the use of AI text detectors to screen submissions [[Knibbs-2023](https://www.wired.com/story/amazon-flag-ai-generated-books/)][[AO3-admin](https://archiveofourown.org/admin_posts/25888)]. As noted in [Knibbs-2023], there are already YouTube tutorials and numerous Reddit threads explaining how to write and publish an ebook quickly, which means that the threat model we describe is not hypothetical. Although humans can sometimes identify AI-generated writing [[Russell-ACL25](https://aclanthology.org/2025.acl-long.267/)], these platforms do not have staff reading and vetting every submission like in traditional publishing houses because the sheer volume of submissions makes manual review infeasible. As a result, these platforms will probably have to rely on automated detectors. Since our results show that current detectors fail to reliably identify Frankentexts, we argue that this technique represents a realistic threat to self-publishing ecosystems and thus call for more robust AI text detectors. We have revised the second to last paragraph of the Introduction section to highlight this argument.

---

> > ### Author Response · Authors · 2025-11-29
> > **[Response 1/3 second follow-up -- Reviewer kEfY] Framing, discussion, and evaluation**
> >
> > **3. [Discussion] The authors have not discussed ethical implications for plagiarism or for AI detectors.**
> >
> > > This work has implications for people using plagiarism / AI-authorship detectors. That's what I'm asking you to discuss, and it's not covered in either your appendix or the updated main text.
> >
> > Because Frankentexts reuse long verbatim spans from human-written sources, using this method to produce “original” fiction for publication would constitute plagiarism in real-world contexts, regardless of whether the collage is assembled by an AI or a human. For this reason, we explicitly do not endorse using our approach to generate or distribute texts intended for public consumption.
> >
> > We have added this discussion to the plagiarism paragraph in the Limitations and Ethical Considerations section.
> >
> > **4. [Evaluation] The paper does provide evidence that Frankentexts are preferred to vanilla texts, but that's not surprising to me.**
> > > I'm not confused, I'm telling you that this result is obvious and intuitive to me. Footnote 7 says that they struggle to reliably follow the instruction. My claim is that a text assembled in this fashion should be obviously preferred to a vanilla generation. These are not the same claim. If you wish to provide evidence against my intuition you could show that, among texts that do follow the instructions generated by a weaker model, those are not preferred to vanilla generations.
> >
> > Thank you for your clarification! We have added Table 9 to the Appendix to show an example where annotators prefer the vanilla generation over the Frankentext produced by Gemini-2.5-Pro. For this pair, an annotator noted that the Frankentext was "a puzzling story that has no consistent plot. Random bits and pieces from elsewhere perhaps?" (this annotation is already included in the final row of Table 3 in the main paper). Even though the human writing snippets selected for this Frankentext are somewhat relevant to the writing prompt, the LLM did not do a good job of connecting them together under the given writing prompt. We believe that this is an example where it is not obvious that Frankentexts are always preferred to vanilla texts.

---

> ### Author Response · Authors · 2025-11-29
> **[Response 2/3 follow up -- Reviewer kEfY] Experiments**
>
> **5. [Experiments] Baseline of retrieval augmented generation where the model retrieves the same texts but isn't instructed to copy them verbatim.**
>
> > Thanks for running this experiment! I agree with your interpretation of the results and am excited to see it included in the paper.
>
> Per the reviewer’s suggestion, we have updated Section 4.1 to include our interpretation of the RAG baseline results.
>
> **6. [Experiments] Whatever the current state-of-the-art in fiction generation should be compared to as well.**
>
> > I don't think you understand my criticism here. I'm saying that, if the primary goal of the paper is to propose a way to write fiction, you should show how this method compares the existing state-of-the-art. Huot et al. (2025) is actually an excellent paper to compare your method to. Without comparisons like this, I can't tell if this is a genuine advance or if it is just something better than the naive approach. Appealing to common usage doesn't countermand this criticism.
>
> > That said, since you've clarified that the primary goal is examining AI detectors and not generating novel fiction this is less important. In this context comparing to how people typically use models makes a lot of sense. I still think that comparisons with other computational methodologies would be interesting, but it's not required.
>
> Thank you for the clarification; we now better understand that the concern is specifically about evaluating our method as a narrative generation technique against other dedicated story generation methods (e.g., Huot et al., 2025), rather than only against frontier LLMs.
>
> We agree that comparisons of Frankentexts with other story generation methods would be helpful for judging Frankentext as a general-purpose narrative generation method. Among methods that build on top of existing frontier models (as in our setup), Huot et al. 2025 is indeed the current state-of-the-art method. However, their method is not yet open-sourced, which makes it difficult for us to include it as our baseline.
>
> In addition, as you said, these methods do not share the same objective as our method, which is to produce high-quality stories that can also evade detection. Therefore, a direct comparison will be irrelevant to the scope of our paper and may not entirely be fair.
>
> We have revised Footnote 8 to reflect this discussion on narrative generation baselines.
>
> **7. [Experiments] The authors fail to compare with light human editing or two models editing each others' work.**
>
> > After reviewing the cited articles I agree that the suggested experiment isn't needed. Please include this information in the paper.
>
> Per the reviewer’s suggestion, we have added this information to the "Other methods for evading AI text detectors" paragraph in the Limitations and Ethical Considerations section, explaining why we opt not to include these detectability baselines.
>
> **8. [Experiments] Although AI-detectors have had success marketing themselves as commercial tools, I do not know any AI researchers who seriously believe that they are reliable in the face of adversarial generations.**
>
> > After reviewing the linked papers I agree that Pangram is very good and withdraw this criticism.
>
> We thank the reviewer for their acknowledgement. We have also included this information in Section 3.3 of the paper to stress the soundness of our detector baseline.
>
> **References**
>
> [Knibbs-2023] Knibbs, Kate. “AI Detection Startups Say Amazon Could Flag AI Books. It Doesn’t.” WIRED, 28 Sept. 2023, www.wired.com/story/amazon-flag-ai-generated-books/. wired.com
>
> [AO3-admin] “Archive of Our Own – Admin Post #25888.” Archive of Our Own, n.d., archiveofourown.org/admin_posts/25888.
>
> [Wang-IJCAI25] Zeng, Zijie, Shiqi Liu, Lele Sha, Zhuang Li, Kaixun Yang, Sannyuya Liu, Dragan Gašević, and Guangliang Chen. “Detecting AI-Generated Sentences in Human-AI Collaborative Hybrid Texts: Challenges, Strategies, and Insights.”Proceedings of the Thirty-Third International Joint Conference on Artificial Intelligence (IJCAI-24), Special Track on AI for Good, 2024.
>
> [Saha-ACL25] Saha, Shoumik, and Soheil Feizi. “Almost AI, Almost Human: The Challenge of Detecting AI-Polished Writing.”Findings of the Association for Computational Linguistics: ACL 2025, 2025.
>
> [Chatterji-NBER25] Chatterji, Aaron, Thomas Cunningham, David Deming, Zoë Hitzig, Christopher Ong, Carl Yan Shan, and Kevin Wadman. “How People Use ChatGPT.” National Bureau of Economic Research Working Paper 34255, 2025.
>
> [He-CHI25] He, Jessica, Stephanie Houde, and Justin D. Weisz. “Which Contributions Deserve Credit? Perceptions of Attribution in Human–AI Co-Creation.” Proceedings of the 2025 CHI Conference on Human Factors in Computing Systems (CHI ’25), 2025.

---

### Author Response · Authors · 2025-11-29
**Rebuttal summary**

Here, we summarize the reviewer’s main concerns and our responses. We have also highlighted additions (blue) and deletions (red) in the updated version of the paper.

**1. [Framing - Reviewer kEfY & 3dau] What is the motivation behind this generation method?**

Our paper is motivated by a simple question: *Can we generate high-quality texts that can fool AI text detectors?* Our findings show that we can. Using Frankentexts, we have exposed vulnerabilities in current detection methods while presenting a non-traditional story generation approach that results in greater creativity and diversity than baseline models.

Frankentexts opens up applications in training data creation and co-writing research, but it also raises a security concern. Bad actors could use this technique to mass-produce undetectable narratives, which poses a threat to self-publishing platforms and the livelihoods of creative professionals. We use our findings to call for more robust AI text detectors.

→ We revised the abstract, introduction, and the methodology section to clarify the framing.

**2. [Framing - Reviewer kEfY] Should Frankentexts be viewed as AI-generated or human-written texts?**

Although different contexts can result in different interpretations, we believe that Frankentexts fall into the hybrid category of mixed human-AI writings.

→ We moved our Ethical Considerations and Limitations section to the main paper and added a paragraph on Authorship.

**3. [Experiments - Reviewer kEfY] What about the RAG baseline where the model retrieves the same texts but isn't instructed to copy them verbatim?**

We introduce a new experiment with this baseline (implemented with Gemini-2.5-Pro and 1,500 relevant paragraphs) and show that it underperforms Frankentexts in both writing quality and detectability. We conclude that this baseline does not contribute to our goal of producing high-quality texts that can fool text detectors.

→ We added this result in Table 2 (main result table) and Section 4.1.

**4. [Experiments - Reviewer 3dau & uMaE] Why is retrieving relevant passages not helpful for the task?**

The model rarely makes full use of the MCP server: it issues 3-5 calls instead of the required 20, which leaves it with too few passages to improve its output. We also find in GPT-5’s reasoning traces that the model stops retrieving and introduces its own content when it has trouble integrating the retrieved paragraphs into the narrative.

→ We moved the agentic implementation results table to the main text (Table 4) and added further explanations in Appendix L.

**5. [Experiments - Reviewer kEfY] What about other detectability baselines, such as light human editing or two models editing each others' work?**

Regarding the baseline where two models edit each others' work, prior work [Russell-ACL25][Masrour-GenAIDetect25] has already shown that Pangram (our evaluation method) is robust to LLM texts "humanized" using o1-pro. As for lightly edited human baselines, this option is costly in time (if done manually) or money (if outsourced), and it cannot be cheaply or quickly automated, which makes it less practical in the context of security risks to writing marketplaces.

→ We added more discussion in the Detectability paragraph of Section 3.3.

**6. [Evaluation - Reviewer uxPN & uMaE] Are writing quality metrics robust to incoherent texts?**

We introduce a new experiment with disjointed texts (Gemini Frankentexts but without any connective texts) and show that, although the distinctness and surprise results for these disjointed texts remain similar to Frankentexts, utility and overall LLM-judge scores drop sharply. Therefore, Frankentexts’ improved writing scores are not merely the result of reused creative phrases.

→ We added the results to Appendix T.

**7. [Discussion – Reviewer uMaE, uxPN & kEfY] What are the ethical considerations surrounding misuses including plagiarism and using copyrighted materials as the source corpus?**

Plagiarism: We do not condone using our method to generate or distribute texts for public consumption. Because Frankentexts copy long verbatim spans from human-written sources, these kinds of texts would constitute plagiarism, whether assembled by an AI or a human.

Human writing datasets: users of Frankentexts are free to use non-copyrighted or self-authored materials for research purposes such as co-writing or detector training. Our use of Books3 is solely to illustrate how malicious actors could misuse the technique to skirt AI-text detectors, not to endorse such use.

Defenses against Frankentexts: We further clarify that our goal is to highlight the existence of adversarial generation methods and to call for more effective attribution tools. Designing defenses, like a detector that is robust to these texts, is thus beyond the scope of this paper.

→ We added paragraphs on Plagiarism, Human writing dataset & Defending against Frankentexts to the Ethical Considerations and Limitations section.

---

### Note · Authors · 2026-01-06

I have read and agree with the venue's withdrawal policy on behalf of myself and my co-authors.